# Design and performance investigation of metamaterial-inspired dual band antenna for WBAN applications

Usman Ali[1], Sadiq Ullah[1]*, Abdul Basir[2], Sen Yan[3], Hongwei Ren[3], Babar Kamal[4], Ladislau Matekovits[5,6,7]*

1 Department of Telecommunication Engineering, University of Engineering & Technology, Mardan, Pakistan, 2 Faculty of Information Technology and Communication Sciences, Tampere University, Tampere, Finland, 3 School of Information and Communications Engineering, Xi'an Jiaotong University, Xi'an, China, 4 Center of Intelligent Acoustics and Immersive Communications, Northwestern Polytechnical University, Xi'an, China, 5 Department of Electronics and Telecommunications, Politecnico di Turin, Turin, Italy, 6 National Research Council of Italy, Istituto di Elettronica e di Ingegneria dell'Informazione e delle Telecomunicazioni, Turin, Italy, 7 Politehnica University Timişoara, Timişoara, Romania

* ladislau.matekovits@polito.it (LM); sadiqullah@uetmardan.edu.pk (SU)

**Data Availability Statement:** Within the manuscript itself

**Funding:** The author(s) received no specific funding for this work.

## Abstract

This paper presents the design and analysis of a metamaterial-based compact dual-band antenna for WBAN applications. The antenna is designed and fabricated on a 0.254 mm thick semi-flexible substrate, RT/Duroid® 5880, with a relative permittivity of 2.2 and a loss tangent of 0.0009. The total dimensions of the antenna are $0.26\lambda_o \times 0.19\lambda_o \times 0.002\lambda_o$, where $\lambda_o$ corresponds to the free space wavelength at 2.45 GHz. To enhance overall performance and isolate the antenna from adverse effects of the human body, it is backed by a 2×2 artificial magnetic conductor (AMC) plane. The total volume of the AMC integrated design is $0.55\lambda_o \times 0.55\lambda_o \times 0.002\lambda_o$. The paper investigates the antenna's performance both with and without AMC integration, considering on- and off-body states, as well as various bending conditions in both *E* and *H*-planes. Results indicate that the AMC-integrated antenna gives improved measured gains of 6.61 dBi and 8.02 dBi, with bandwidths of 10.12% and 7.43% at 2.45 GHz and 5.80 GHz, respectively. Furthermore, the AMC integrated antenna reduces the specific absorption rate (SAR) to (>96%) and (>93%) at 2.45 GHz and 5.80 GHz, meeting FCC requirements for low SAR at both frequencies when placed in proximity to the human body. CST Microwave Studio (MWS) and Ansys High-Frequency Structure Simulation (HFSS), both full-wave simulation tools, are utilized to evaluate the antenna's performance and to characterize the AMC unit cell. The simulated and tested results are in mutual agreement. Due to its low profile, high gain, adequate bandwidth, low SAR values, and compact size, the AMC integrated antenna is considered suitable for WBAN applications.

## Introduction

Wearable wireless devices are increasingly vital in advanced communication systems like the Internet of Things (IoTs) and fifth-generation (5G) technology [1–3]. These devices serve

**Competing interests:** The authors have declared that no competing interests exist.

various body-worn applications, including medical monitoring, sports training, emergency services, health monitoring, military communication systems, brain control, and interaction, as illustrated in Fig 1. Antennas in these systems play a crucial role as the front-end component, determining overall system efficiency [4]. Unlike conventional antennas operating in free space, wearable antennas are positioned near human body tissues. These tissues, with complex permittivity and conductivity, impact antenna performance metrics such as reflection coefficient, gain, directivity, impedance bandwidth, and radiation characteristics [5]. Therefore, these antennas must maintain high gain, acceptable efficiency, and stable reflection coefficients to ensure effective communication. Moreover, when antennas operate near the human body, they emit radiation not only in the desired direction but also towards the body. This unintended radiation, when absorbed by body tissues, can potentially harm them, leading to serious health implications. Hence, safety considerations, measured by the specific absorption rate (SAR), are crucial [6]. The SAR, measured in watts per kilogram (W/kg), quantifies the energy absorbed by the body as heat due to interaction with electromagnetic radiation. SAR values are typically averaged over specific volumes of human body tissues, such as 1 gram or 10 grams. It is important to ensure SAR remains within safety thresholds set by American (US) and European (EU) standards. According to US standards, the SAR limit should not exceed 1.6 W/kg for every 1 gram of human body tissue, while EU standards set the limit at 2 W/kg for every 10 grams of tissue [7]. The SAR can be calculated using the general mathematical equation presented in [8].

$$SAR = \frac{\sigma |E|^2}{\rho} \, [\text{W/Kg}] \tag{1}$$

Where, the electric field strength, denoted as $E$ in units of V/m, represents the intensity of the electric field. The conductivity of the tissue, expressed as $\sigma$ in units of S/m, refers to its ability to conduct electric current. The mass density of the tissue, denoted by $\rho$ in units of kg/m$^3$, represents its mass per unit volume.

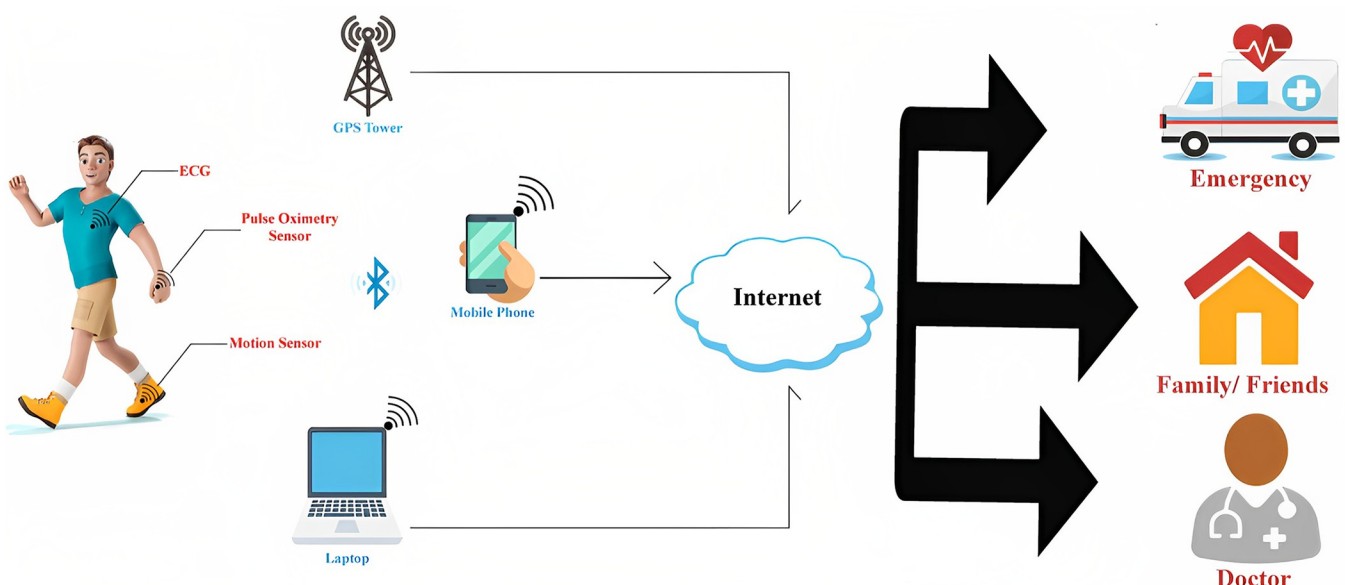

**Fig 1. Generic model of medical care using wearable electronics [3].**

Additionally, these antennas need to be flexible, low-profile, and miniaturized to fulfill the requirements of Wireless Body Area Network (WBAN) systems [9–12]. Therefore, special attention should also be given to the overall size of the antenna, aiming for a compact design [13]. Moreover, these antennas are subject to various deformations such as crumpling, stretching, and bending, which can alter their overall performance in terms of gain, efficiency, frequency detuning, and driving point impedance [14, 15]. Consequently, designing these antennas poses significant challenges in achieving high gain and efficiency, as well as ensuring flexibility, conformability, and compliance with safety regulations regarding radiation exposure to human body tissues. For wearable devices to gain acceptance in the industry, the antennas used in these devices must exhibit high gain and efficiency, low SAR values, and flexibility that aligns with the targeted applications [16].

In the literature, researchers have proposed various wearable antenna designs for numerous body-worn applications. For example, different types of wearable antennas operating in single and dual frequency bands have been proposed [17]. However, depending on the geometry of the proposed designs, frequency detuning can occur on either side of the resonant frequency, adversely affecting on-body gain and radiation efficiency. Additionally, the SAR value of these antennas may exceed the limits specified by regulatory bodies. To address these issues, researchers have proposed various techniques. For instance, wideband and ultra-wideband (UWB) antennas have been designed on wearable, textile, or flexible materials to mitigate frequency detuning, as reported in [18, 19]. However, these antennas encounter interference issues.

To improve overall performance and address issues such as frequency detuning, back lobe radiation, coupling with lossy human body tissues, and SAR reduction to meet safety limits, researchers have proposed and employed various techniques [20–22]. However, these techniques have limitations. Recently, metamaterials (MTMs) have gained traction as backing shields for body-worn antennas to enhance performance, increase isolation from the body, reduce SAR, and stabilize reflection coefficients. MTMs, also known as electromagnetic bandgap (EBG), artificial magnetic conductor (AMC), and high impedance surface (HIS), exhibit unique electromagnetic characteristics. Their applications in body-worn antenna design are detailed in [17, 23–25]. Incorporating these materials into antennas suppresses surface waves, reducing backward radiation towards the body and improving overall performance, including increased Front-to-Back Ratio (FBR) and decreased SAR values [26–28]. Moreover, they minimize frequency detuning when antennas are worn on the body [29, 30]. Different metamaterials have been used as ground planes to enhance radiation characteristics, as discussed in [24, 31]. Various AMC structures have also been integrated to manipulate wave fronts and reduce SAR [12, 32–34]. Additionally, metamaterial surfaces have been utilized for energy harvesting in body-worn scenarios; for example [35], proposed integrating Electromagnetic Bandgap (EBG) structures into wearable microstrip antennas for improved performance and RF energy harvesting. Furthermore [36], presented a miniaturized antenna-based wearable self-powered system, exploring the integration of a compact, flexible Solant-Rectenna with low-energy devices. EBG structures in this system serve as a ground plane, mitigating losses and shielding the body from unwanted electromagnetic radiation leakage. To achieve required flexibility and conformability, researchers have explored various antenna designs on flexible/semi-flexible materials such as polyimides [37–41], papers [42, 43], polyester films, latex, and textiles [17, 44].

In summary, to meet the requirements of wearable antennas, such as high gain, adequate bandwidth, efficiency, protection against human body effects, mitigation of back lobe radiation to lower SAR values, and flexibility, the proposed work introduces a compact, metamaterial-inspired dual-band wearable antenna operating at 2.45 GHz and 5.8 GHz ISM bands on a semi-flexible substrate. The antenna aims to fulfill the criteria of high gain, acceptable efficiency, and protection against human body effects, low SAR values, and flexibility. Integration

of an artificial magnetic conductor (AMC) structure addresses performance degradation, minimizes SAR, and isolates the human body from backward radiation when the antenna is in close proximity. This integration enhances antenna performance and effectively controls SAR levels. The design and analysis of the proposed work are conducted using CST Microwave Studio and High-Frequency Structure Simulation (HFSS), both full-wave simulation tools. The fabricated prototype is analyzed and tested, and the experimental results are validated. The improved performance and reduced SAR, as observed in the tested results, render it suitable for various wearable applications.

The remaining paper is organized as follows: Section 2 covers the proposed antenna design, its parametric study, and the in-phase characterization of metamaterial. Section 3 analyzes, compares, and discusses the results obtained from the on-body, off-body, and bending investigations of the proposed antenna with and without metamaterial. SAR analysis is explored in Section 4. Finally, Section 5 concludes the work and recommends some future directions.

## Design and MTM characterization

In this section, the CST MWS/HFSS simulation tools, based on the Finite Integration Technique (FIT) were utilized, to design and analyze the proposed wearable antenna and the associated MTM structure. The design process commences with the conventional method for designing the dual-band antenna. Subsequently, the design and characterization of the dual-band MTM, operating at the desired frequency bands of 2.45 GHz and 5.8 GHz was conducted. In the subsequent sections, the integration of the MTM with the antenna and the on- and off-body performance analysis were investigated.

### Antenna topology and evolution process

The antenna design was initially based on the traditional edge-feed rectangular microstrip patch antenna design method [45]. The rectangular microstrip patch was optimized to achieve the desired frequency bands. The geometry of the proposed antenna, operating at dual frequency bands of 2.45 GHz and 5.8 GHz, is depicted in Fig 2. The antenna was constructed using a 0.254 mm thick flexible RT/Duroid® 5880 material with a relative permittivity of 2.2 and a loss tangent of 0.0009. The overall volume of the antenna is $0.26\lambda_o \times 0.19\lambda_o \times 0.002\lambda_o$, where $\lambda_o$ signifies the free space wavelength at the lowest frequency (2.45 GHz). The evolution of the proposed design is a three-step process, as shown in Fig 3(A), with the corresponding reflection coefficients in dB presented in Fig 3(B). The current path distribution was utilized as a design methodology.

At *Step 1*, the initial design comprised a simple edge-fed rectangular patch using a 50 Ω microstrip transmission line. It exhibited a single frequency band centered at 3.26 GHz with a good reflection coefficient ($|S_{11}|$) of less than -14 dB and a -10 dB bandwidth of 480 MHz (3.04 GHz—3.52 GHz). In *Step 2*, to achieve dual-band characteristics, two horizontal *L*-shaped stubs of the same dimensions were added to the left and right sides of the topmost part of the rectangle. This modification introduced two distinct frequency bands. The first band centered at 2.67 GHz exhibited a reflection coefficient ($|S_{11}|$) of less than -14 dB and a -10 dB bandwidth of 330 MHz (2.60 GHz—2.93 GHz). The second band centered at 6.21 GHz had a reflection coefficient ($|S_{11}|$) of less than -51 dB and a -10 dB bandwidth of 430 MHz (6.01 GHz—6.44 GHz). While dual-band characteristics were achieved at this step, however, the resonances were undesirable, and impedance matching needed improvement.

To achieve the desirable frequency bands, in *Step 3*, the electrical length of the radiating structure was modified by introducing three cuts of equal dimensions ($L_2 = L_5 = L_6 = 8$ mm). These cuts led to the desired resonance frequencies: the first band centered at 2.45 GHz with a

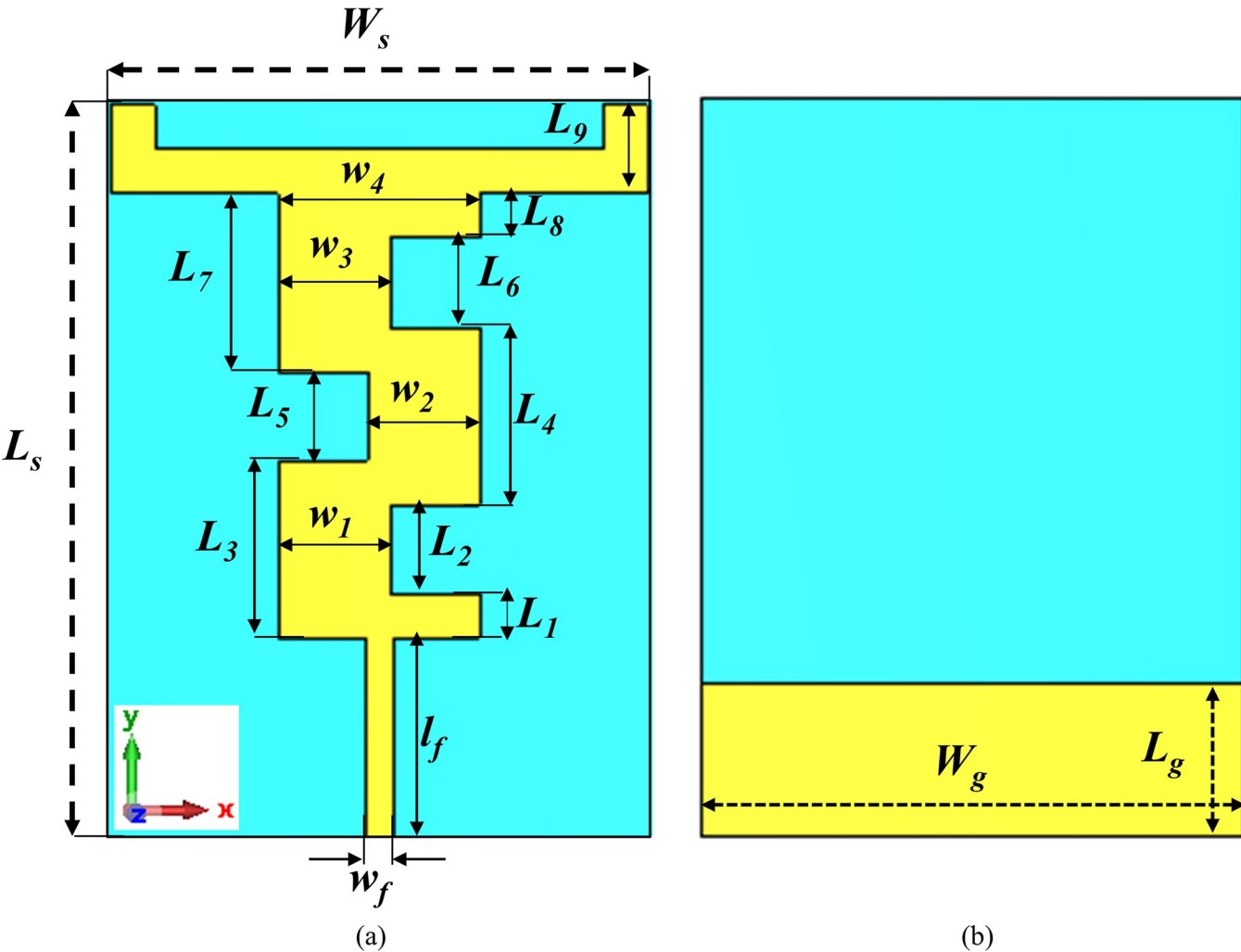

**Fig 2.** Topology of the proposed antenna (a) front view (b) back view.

reflection coefficient ($|S_{11}|$) of less than -19 dB and a -10 dB bandwidth of 270 MHz (2.33 GHz—2.60 GHz), and the second band centered at 5.80 GHz with a reflection coefficient ($|S_{11}|$) of less than -26 dB and a -10 dB bandwidth of 340 MHz (5.65 GHz—5.99 GHz). The optimum dimensions of the proposed dual-band design achieved in Step 3 are summarized in Table 1. The prototype of the proposed design was fabricated using a PCB machine, and a 50 Ω standard SMA connector was used for the excitation.

The radiation mechanism of the proposed antenna at two distinct frequencies is further confirmed by the density of surface currents. Surface current density, which shows the flow of currents across the geometry of the antenna, is a critical factor in determining resonance. In Fig 4(A) and 4(B), the distribution of surface currents illustrate how the current flows across the antenna's structure at 2.45 GHz and 5.80 GHz, respectively. These variations in current density contribute to the ability of antenna to resonate effectively and perform efficiently across different frequency bands when subjected to the same excitation at the input port. In Fig 4(A), it is evident that, including some portion of the horizontal L-shaped stubs, the main radiator carries the majority of the current. Specifically, at the lower frequency band of 2.45 GHz, the current is observed to be distributed along the entire length of the main radiator.

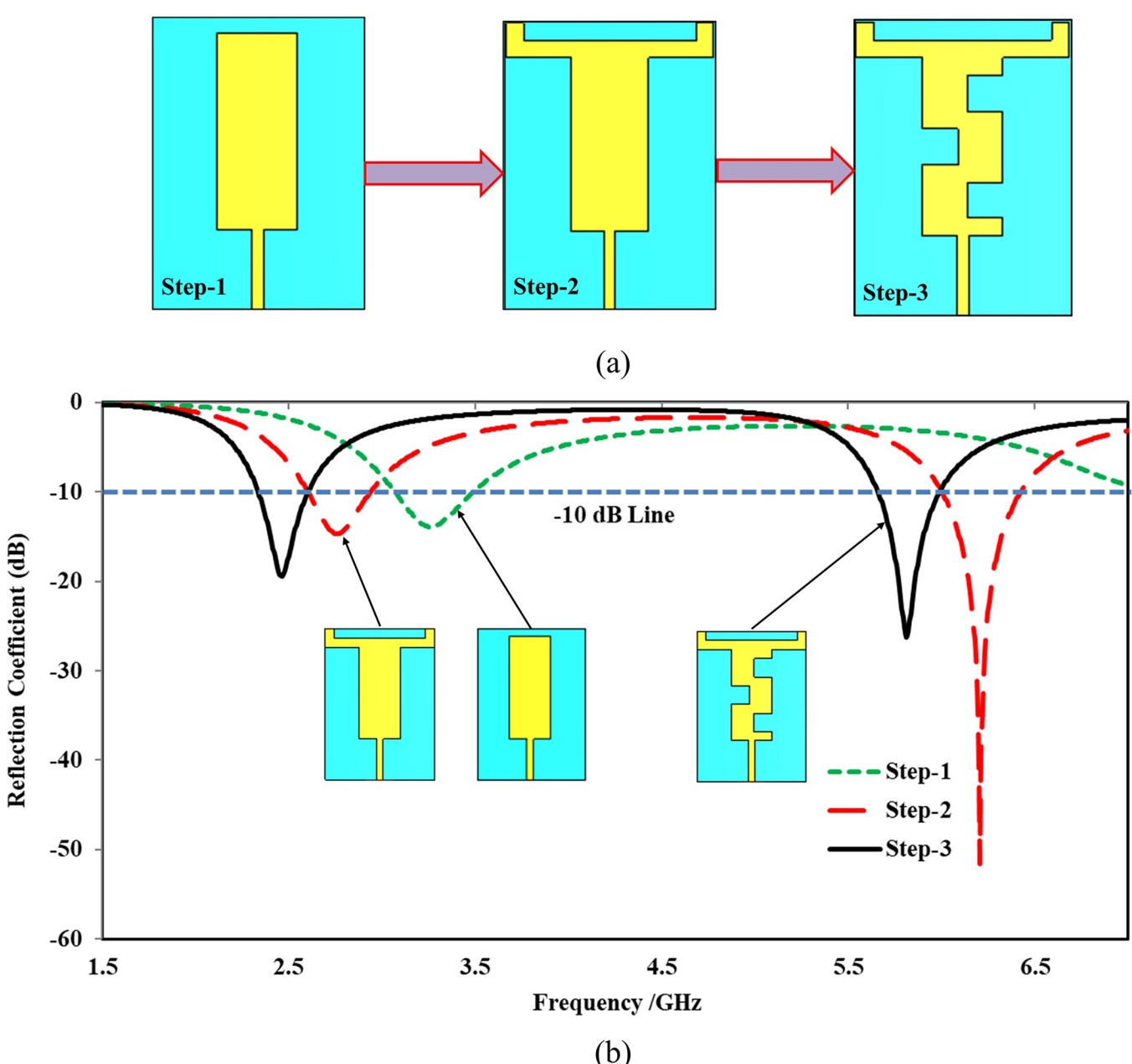

**Fig 3.** Evolution process of the proposed antenna (a) design steps (b) corresponding reflection coefficients attained at various steps.

**Table 1. Summary of the optimized dimensions (mm) of the proposed design.**

| Parameter | Value | Parameter | Value | Parameter | Value | Parameter | Value |
|---|---|---|---|---|---|---|---|
| $L_s$ | 33 | $L_2$ | 4.0 | $L_7$ | 8.0 | $w_2$ | 5.31 |
| $W_s$ | 24 | $L_3$ | 8.0 | $L_8$ | 2.0 | $w_3$ | 5.31 |
| $w_f$ | 1.3 | $L_4$ | 8.0 | $L_9$ | 4.0 | $w_4$ | 9.32 |
| $l_f$ | 9.0 | $L_5$ | 4.0 | $L_g$ | 7.0 | -- | -- |
| $L_1$ | 2.0 | $L_6$ | 4.0 | $w_1$ | 5.31 | -- | -- |

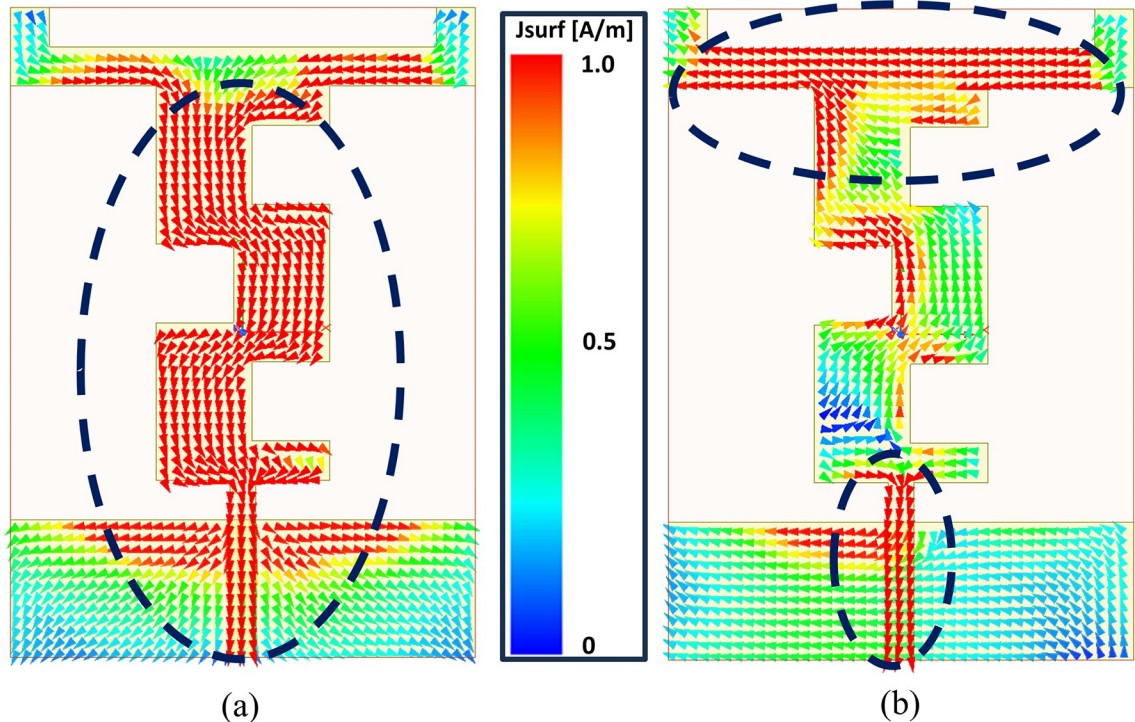

**Fig 4.** Surface currents density attained in the final step at (a) 2.45 GHz (b) 5.80 GHz.

However, at the higher frequency band of 5.80 GHz, as depicted in Fig 4(B), the current distribution pattern shifts slightly. Here, the current predominantly concentrates on three main segments: the bottom section of the feedline, a specific portion of the main radiator where the upper cut ($L_5$) is introduced, and the upper part of the main radiator, including a segment of the horizontal L-shaped stub.

This observation highlights the distinct radiation mechanisms employed by the antenna at the two operating frequencies. The concentration of current at different segments of the antenna indicates the specific regions that mainly contribute to radiation at each frequency band. Overall, this analysis offers valuable perspectives on how the antenna effectively radiates at both 2.45 GHz and 5.80 GHz frequencies, facilitating into a more understanding of its performance characteristics.

## Design and in-phase characterization of MTM

The perfect electrical conductor (PEC) reflects plane waves incident normal to its plane with a 180-degree phase shift, while the perfect magnetic conductor (PMC) reflects them with a 0-degree phase shift. According to image theory, when a PEC ground plane is used, the image current and the antenna's current cancel each other, leading to a decrease in the real part of the impedance approaching 0Ω. Simultaneously, the imaginary part of the impedance increases towards infinity. Consequently, a significant amount of electromagnetic energy is trapped between the ground plane and the antenna, resulting in a substantial decrease in antenna efficiency [46]. Conversely, when an artificial magnetic conductor (AMC) is employed as the ground plane instead of a PEC, the scenario is reversed due to the in-phase reflection characteristics of the AMC. Thus, using an AMC as a ground plane significantly improves the overall performance of monopole antennas in terms of gain and radiation

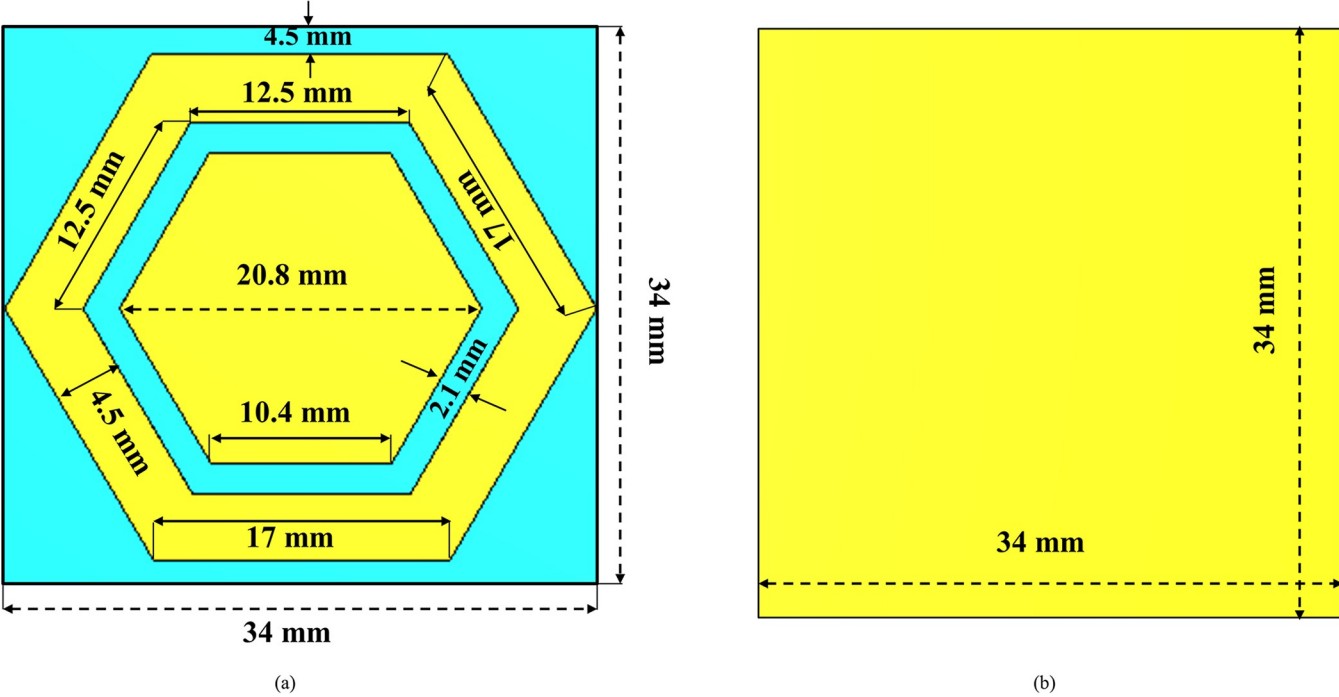

**Fig 5.** Layout of the proposed AMC unit cell with optimized dimensions (a) front side (b) back side.

efficiency. Moreover, in body-worn applications, AMC ground planes are utilized to minimize the coupling between the antenna and human body tissue, resulting in a reduction in SAR value that safeguards the human body tissues from the adverse effects of radiation directed towards them [16, 47]. Furthermore, AMC surfaces are employed to mitigate the impedance mismatch of the antenna caused by the proximity of human body tissues. In this study, a dual-band, hexagonal-shaped MTM structure acting as an AMC was designed and analyzed for in-phase characteristics. The AMC array was then incorporated into the proposed dual-band antenna, and its on- and off-body performance was evaluated. The AMC structure was fabricated on a 0.254 mm thick, semi-flexible RT/Duroid® 5880 material having a relative permittivity of 2.2 and a loss tangent of 0.0009. The structure consisted of an inner hexagonal patch, which is surrounded by the outer hexagonal loop and is separated by 2.1 mm.

The proposed AMC structure demonstrated unique electromagnetic properties at two distinct frequency bands centered around 2.45 GHz and 5.80 GHz. The outer hexagonal loop mainly determines the lower resonant frequency (2.45 GHz), while the inner hexagonal patch determines the higher resonant frequency (5.80 GHz). The dimensions of the hexagonal unit cell were determined using the equations outlined in reference [48]. The volume of the unit cell was $0.27\lambda_o \times 0.27\lambda_o \times 0.002\lambda_o$. Fig 5 illustrates the layout of the optimized AMC unit cell design.

The proposed unit cell was characterized using the higher frequency structure simulation tool. A linearly polarized plane wave ($TE_{10}$) was incident on the surface from the top, directed towards the negative $z$-axis. The boundary conditions along the $x$ and $y$ boundaries were set to the 'unit cell' option. Fig 6 illustrates the simulation setup used for the in-phase characterization of the unit cell. The primary objective was to achieve a reflection phase of 0 degrees at two specific frequencies: 2.45 GHz and 5.80 GHz. The reflection phase of the unit cell at these frequencies, displaying the desired 0-degree phase, is depicted in Fig 7.

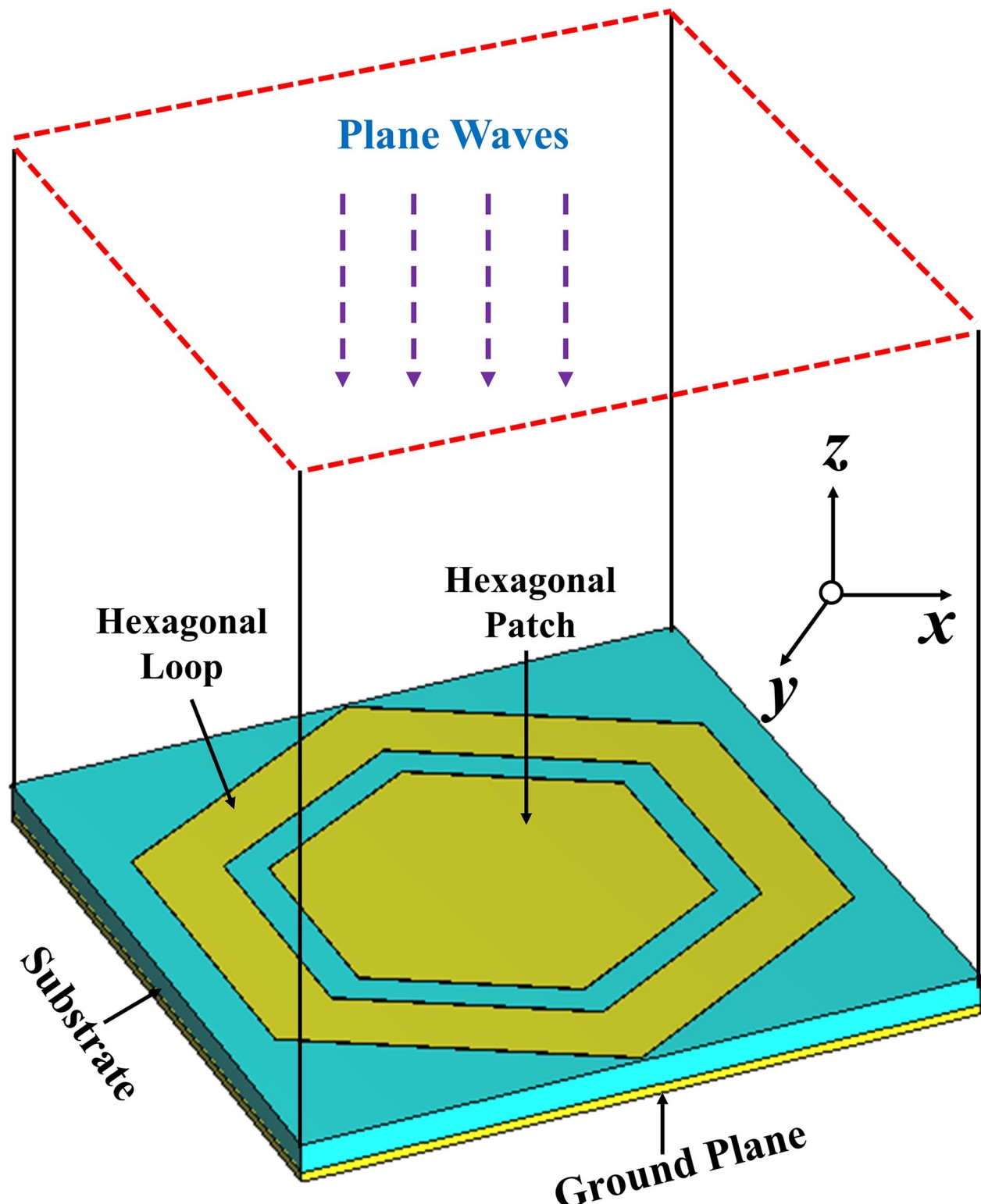

**Fig 6. Simulation setup for in-phase characterization of the proposed unit cell.**

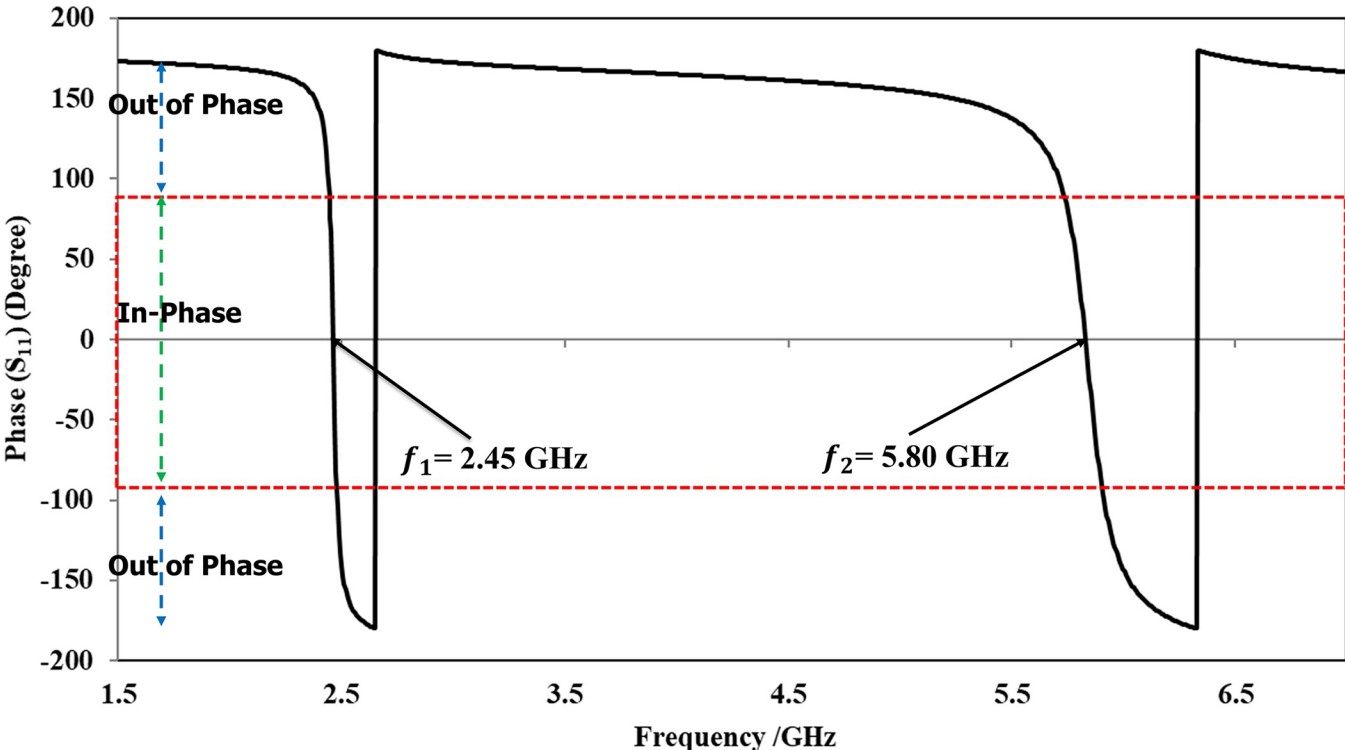

**Fig 7. Simulated reflection phase of the unit cell displays a 0 degree phase at desired frequencies.**

It was observed that the proposed unit cell demonstrated a reflection phase of 0 degrees at the designated frequencies, effectively functioning as a dual-band reflector for the intended dual-band antenna. Specifically, at 2.45 GHz and 5.80 GHz, the hexagonal unit cell structure exhibited similar behavior to a PMC [47]. Within the range of ±180˚ to ±90˚, the structure

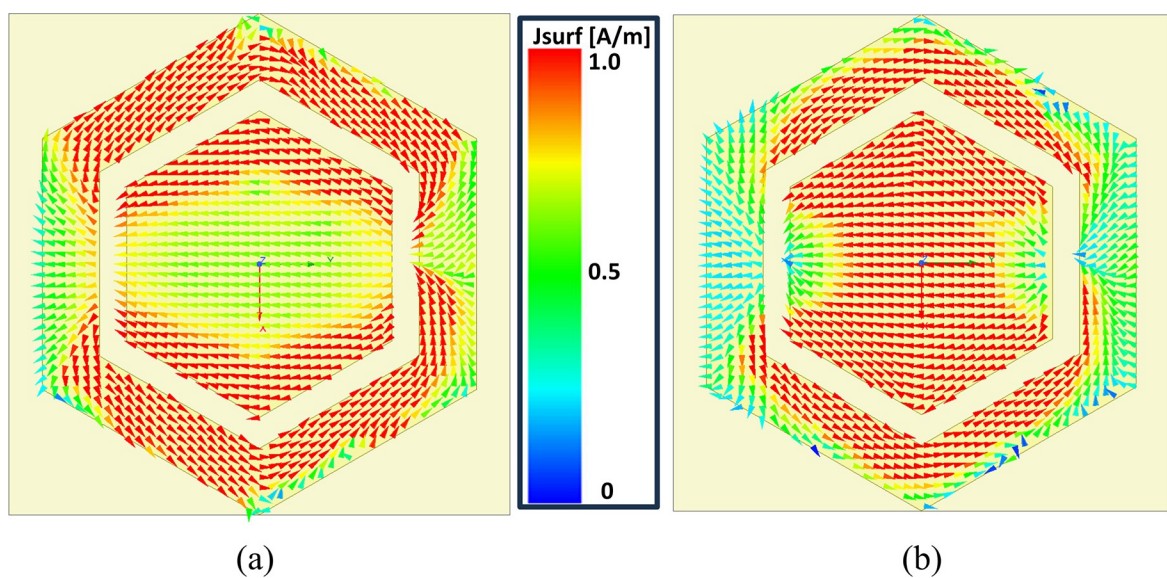

**Fig 8.** Surface currents density of the proposed AMC unit cell at (a) 2.45 GHz (b) 5.80 GHz.

behaved as a Perfect Electric Conductor (PEC). However, in the desired bandwidth, the reflection phase varied from +90 degrees to -90 degrees, crossing the 0-degree reflection phase at the specified frequencies. Within this bandwidth, the proposed hexagonal structure acted as an AMC. This range of frequencies is referred to as the reflection phase bandwidth, which can be calculated using the following equation.

$$BW_{rp} = \frac{f_{-90} - f_{+90}}{f_c} \times 100 \tag{2}$$

Where, $BW_{rp}$ represents the reflection phase bandwidth and $f_{-90}$ and $f_{+90}$ designate the reflection phase at -90$^o$ and +90$^o$, respectively. The proposed AMC unit cell gives a reflection phase bandwidth of 6.64% at 2.45 GHz and 11.9% at 5.80 GHz.

The capacitance of the two hexagons, being of different dimensions, varies accordingly. This disparity allows the proposed design to resonate at two distinct frequencies. These frequencies can be determined using the following relationship:

$$f_1 = \frac{1}{2\pi\sqrt{LC_1}} \tag{3}$$

$$f_2 = \frac{1}{2\pi\sqrt{LC_2}} \tag{4}$$

The observation is supported by evaluating the distribution of surface currents within the proposed hexagonal unit cell. Fig 8(A) and 8(B) illustrate the captured surface currents at 2.45 GHz and 5.80 GHz, respectively. In the proposed unit cell, it is shown that the surface current density at 2.45 GHz is mainly restricted to the outer hexagonal loop, which serves as the primary resonator and is larger in size, while the current at 5.8 GHz primarily lies in the inner hexagonal patch, functioning as the secondary resonator. At the lower frequency band centered at 2.45 GHz, as illustrated in Fig 8(A), the surface currents predominantly circulate around the outer hexagonal loop. This concentrated flow significantly contributes to the 0-degree phase of the unit cell at this frequency. Conversely, at 5.80 GHz, depicted in Fig 8(B), the surface currents mainly concentrate on the inner hexagonal patch. This distribution highlights its role in ensuring a 0-degree phase at this frequency for the proposed unit cell. It can be concluded that the outer hexagonal loop surrounds the inner hexagonal patch to make the structure resonate at the lower frequency, and the inner hexagonal patch has a significant impact on the higher resonant frequency.

## Results and discussion

This section presents a comparison of the simulated and measured performance of the proposed dual-band antenna, both with and without employing an AMC structure, in both on-body and off-body states.

### Off-body analysis without AMC in flat state

The proposed antenna was fabricated and tested to validate the simulated results. Fig 9 shows photographs of the fabricated prototype of the proposed design and the setup using a Vector Network Analyzer (VNA) for analyzing and testing the reflection coefficients. For measuring the far-field gain pattern of the proposed antenna, the setup depicted in Fig 18 was utilized. In the gain measurement process, the antenna was positioned on a positioner in the far field of a broadband horn antenna.

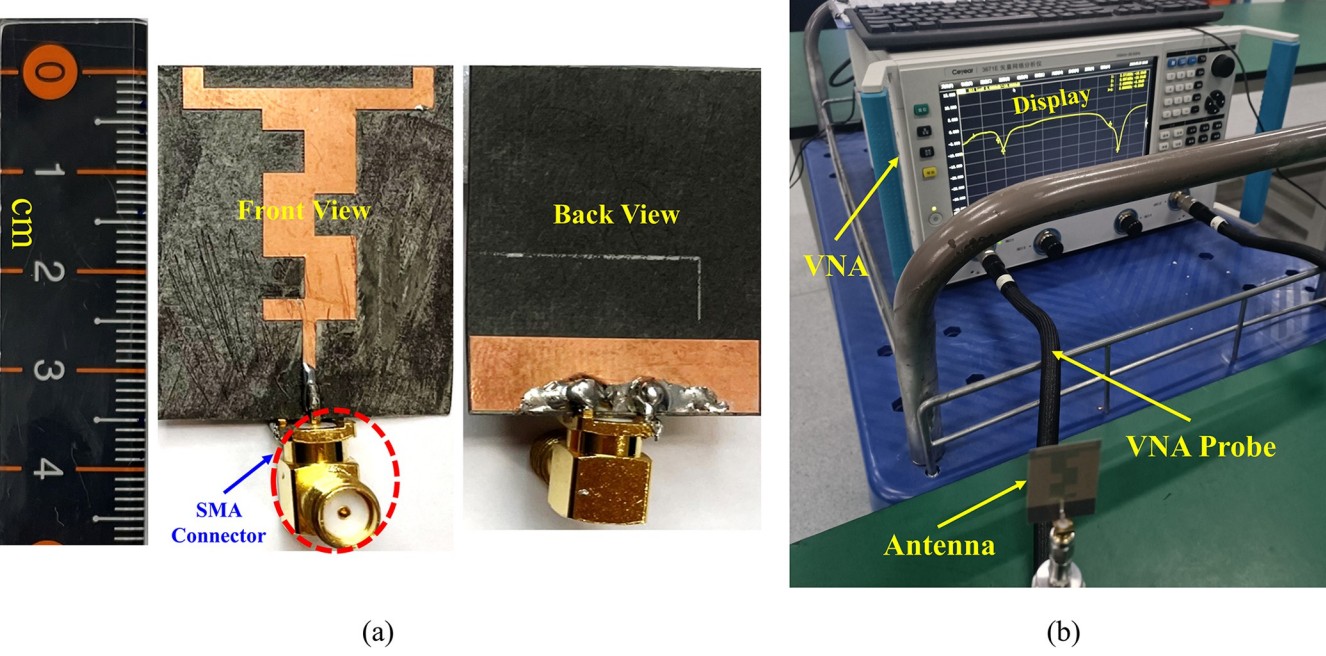

(a)                                                                 (b)

**Fig 9.** Fabricated prototype (b) setup for reflection coefficient measurement.

Fig 10 displays a comparison between the simulated and measured reflection coefficients of the proposed antenna in the off-body state in flat condition. The simulation illustrates that the antenna resonates at 2.45 and 5.80 GHz with ($|S_{11}|$) < -10 dB, providing an impedance bandwidth of 270 and 340 MHz, respectively. The measured results validate the simulated outcomes, indicating that the proposed antenna operates at two distinct frequency bands centered at 2.43 and 5.82 GHz, with an impedance bandwidth of 280 and 385 MHz, respectively. Both the simulated and measured results are well-matched, with only minimal alterations in both frequency bands due to fabrication and soldering effects. However, the entire ISM band is fully covered.

The far-field gain pattern of the proposed design in the off-body flat condition was examined in the E and H planes using the far-field measurement setup in an anechoic chamber, as shown in Fig 18. The comparison between the simulated and tested gain patterns at the desired frequency bands centered at 2.45 and 5.80 GHz is displayed in Fig 11. The peak gain of the proposed antenna at 2.45 and 5.80 GHz is 1.89 and 4.10 dBi, respectively. The radiation pattern in the E-plane resembles a 'figure of eight', while it is omni-directional in the H-plane at the two distinct frequency bands centered at 2.45 and 5.80 GHz, as shown in Fig 11(A) and 11(B), respectively.

The results indicate that the simulated and tested gain patterns are almost identical, further validating that the fabricated antenna provides similar results to those obtained in simulation. The summary of the proposed dual-band antenna in the off-body flat condition is presented in Table 2. For further clarity, the simulated three-dimensional gain patterns at 2.45 and 5.80 GHz frequencies are depicted in Fig 12(A) and 12(B), respectively.

## Off-body analysis without AMC in bent state

Wearable antennas are designed to be worn on the human body in various positions according to the required applications. Due to different human body postures, these antennas can

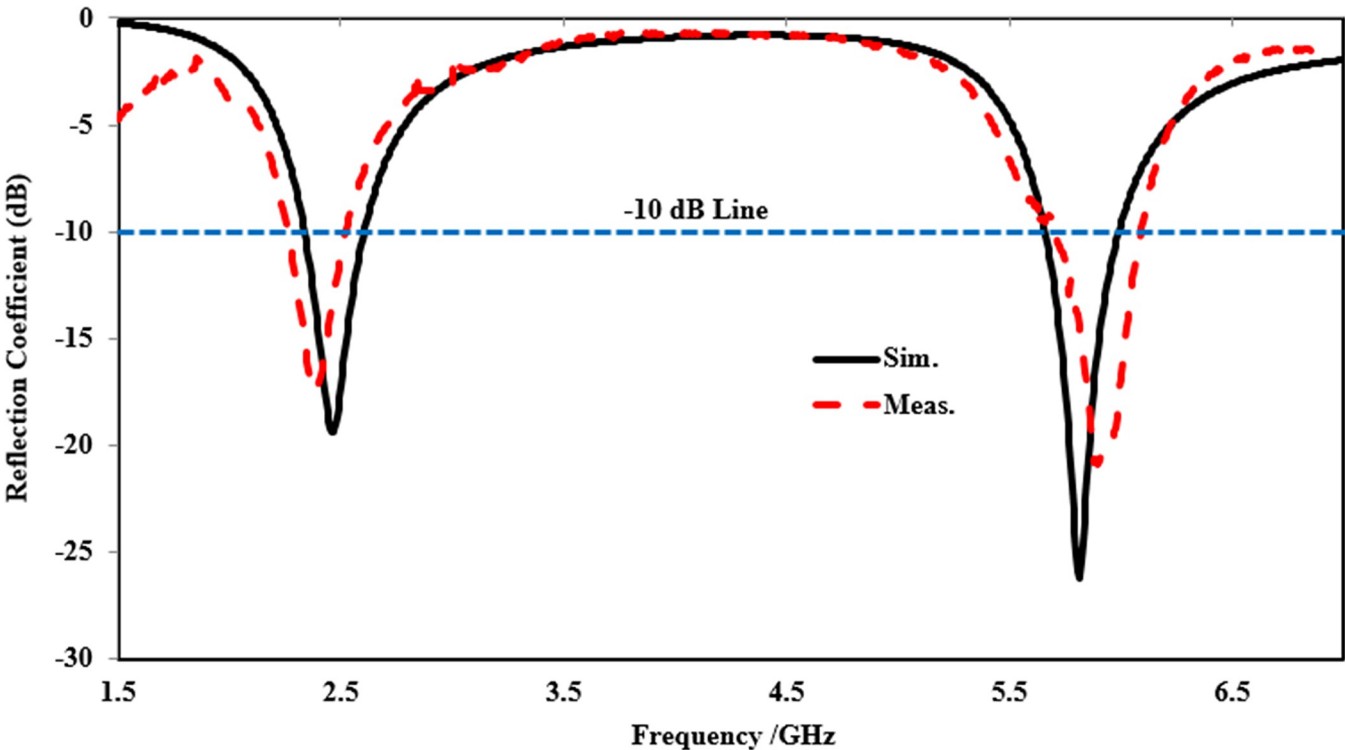

**Fig 10. Reflection coefficient comparison of the proposed antenna in off-body flat state.**

experience physical deformation. Therefore, it is necessary to evaluate the performance of these antennas under various deformations such as bending, stretching, and crumpling. In this section, the performance of the proposed antenna was evaluated in its off-body state under various bending angles, including 15°, 30°, and 45° in the *E* and *H*-planes, as illustrated in Fig 13. The bending process was conducted in CST MWS using vacuum cylinders of corresponding diameters. The bending angles were chosen based on the sizes of adult human leg and arm models

The off-body comparison of the simulated reflection coefficients of the proposed antenna in flat and bent conditions in both planes is portrayed in Fig 14(A) and 14(B), respectively. When the antenna was bent in the *E*-plane, the reflection coefficient remained quite stable even when bent at an angle of 45°, as illustrated in Fig 14(A). However, a slight reduction in the reflection coefficient was observed at the lower band centered at 2.45 GHz. This reduction was more pronounced when the antenna was bent at 45° compared to 30° and 15° of bending. Specifically, the reflection coefficient of the bent antenna was reduced from -19 dB (flat state) to -17 dB (bent state). Additionally, an improvement in the reflection coefficient was noticed at the higher band (i.e., 5.80 GHz). This enhancement in the reflection coefficient was significant when the antenna was bent at 45° compared to 30° and 15°, with the reflection coefficient of the bent antenna enhanced from -25 dB (flat state) to -38 dB (bent state). However, the effect of bending in the *E*-plane at various angles was insignificant and considered negligible.

Similarly, when the antenna was bent in the *H*-plane at angles of 15°, 30°, and 45°, slight frequency detuning in the lower frequency band centered at 2.45 GHz was observed at all three bending angles. This frequency detuning was greater at a smaller degree of bending (i.e., 15°) compared to the other two bending angles. Additionally, a reduction in the reflection coefficient was observed at both frequency bands for all bending angles, with a greater

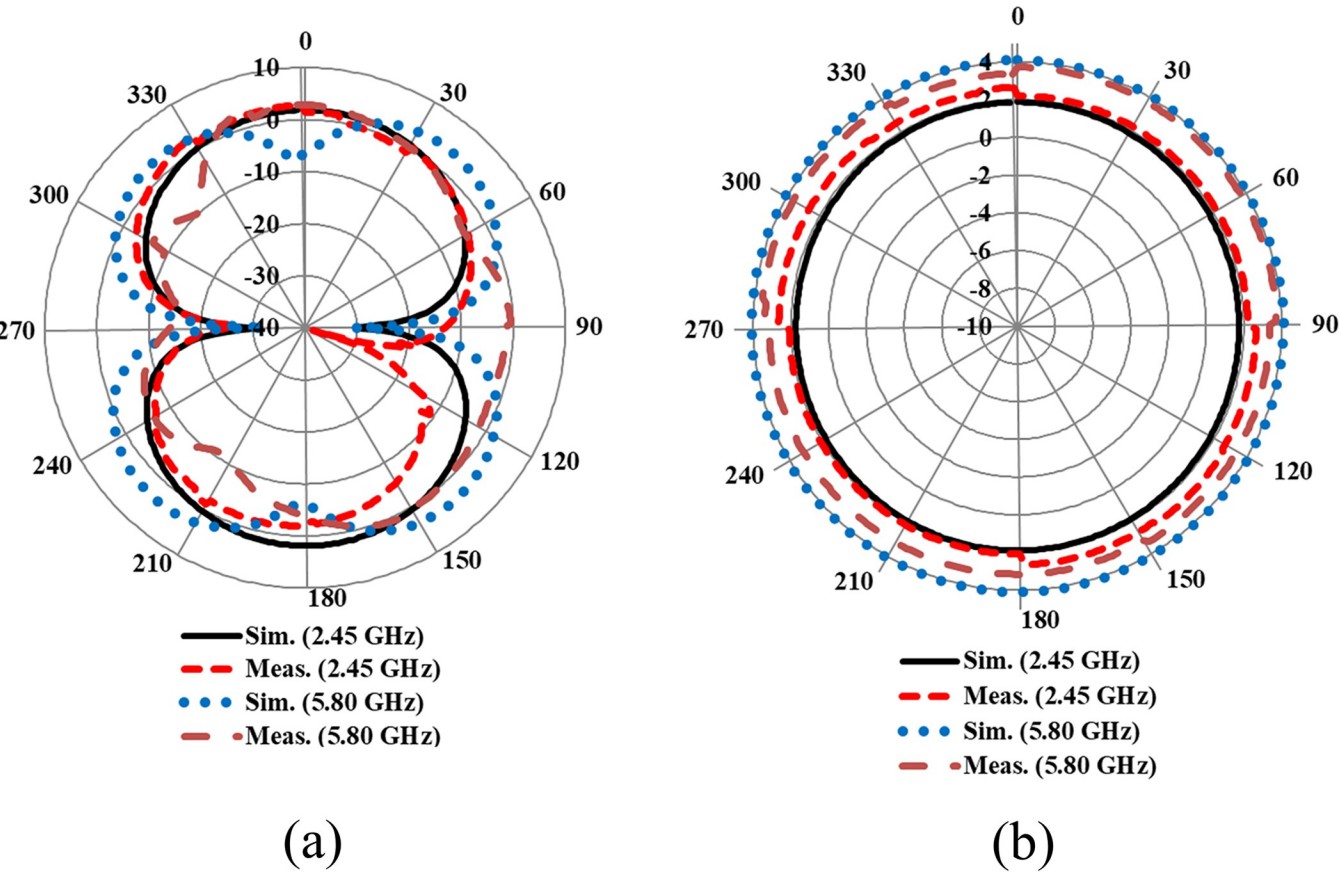

**Fig 11.** Far-field gain comparison of the proposed antenna in (a) *E*-plane (b) *H*-plane.

reduction at 15˚ of bending compared to 45˚ and 30˚. The variation in the reflection coefficient is attributed to the change in the geometry of the antenna, which affects its electrical properties, consequently impacting the propagation of electromagnetic waves and leading to changes in impedance matching and, subsequently, the reflection coefficient. Moreover, the bandwidth also fluctuates; however, the entire ISM band is still covered. In general, regardless of the bending angle and planes, the proposed antenna exhibited stable performance, providing adequate bandwidth coverage of the ISM band. A summary of the performance of the proposed antenna under various bending angles in both principal planes is given in Table 3.

### Off-body analysis with AMC-integration in flat state

The conventional dual-band wearable antenna discussed in the previous section was integrated with a 2×2 AMC array, with an overall size of 68×68 mm$^2$, corresponding to $0.55\lambda_o \times 0.55\lambda_o$.

**Table 2. Summary of the performance comparison of the proposed antenna in off body flat state.**

| Parameters | @ 2.45 GHz | | @ 5.80 GHz | |
|---|---|---|---|---|
| | Sim. | Meas. | Sim. | Meas. |
| Gain (dBi) | 1.89 | 2.07 | 4.10 | 3.12 |
| VSWR | 1.13 | 1.12 | 1.11 | 1.17 |
| -10 dB Bandwidth (MHz) | 270 | 279 | 340 | 363 |
| Radiation efficiency (%) | 96.44 | 87.29 | 92.48 | 84.12 |

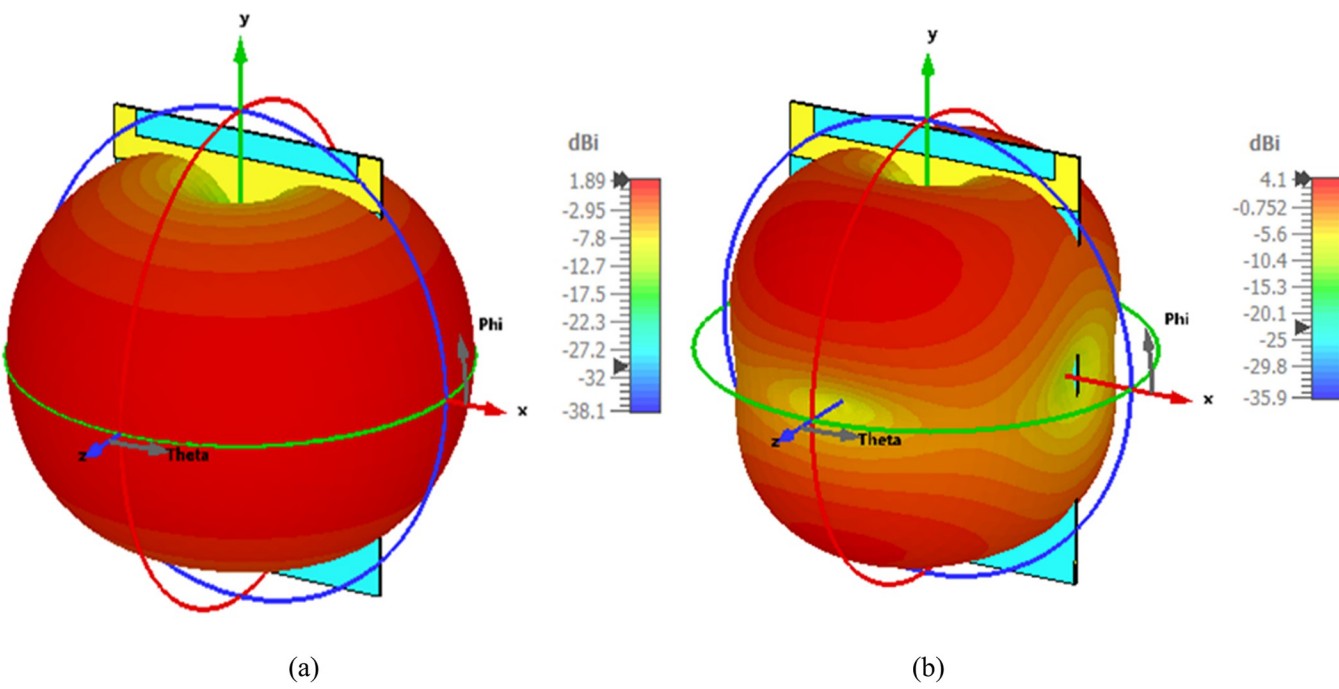

(a)                                                    (b)

**Fig 12.** Simulated three dimensional (3D) gain pattern at (a) 2.45 GHz (b) 5.80 GHz.

The arrangement of the proposed antenna integrated with the AMC structure is illustrated in Fig 15. The antenna is positioned above the 2×2 AMC array at a certain distance. In the fabricated prototype, commercially available Styrofoam is used between the AMC array and the antenna to reduce coupling and avoid short circuits and interaction with the SMA connector.

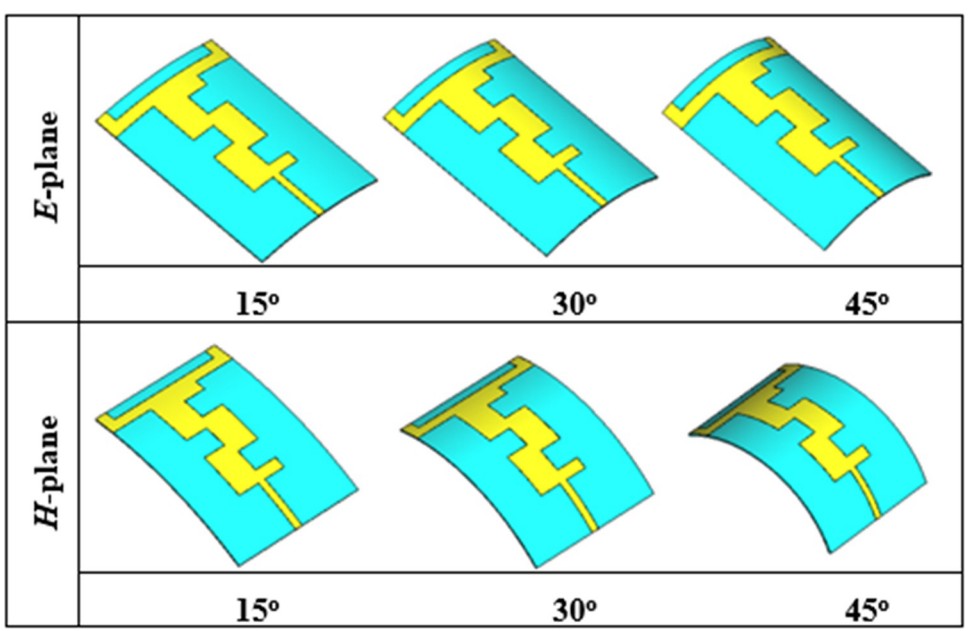

**Fig 13. Layout of the bent antenna at various angles in both principal planes.**

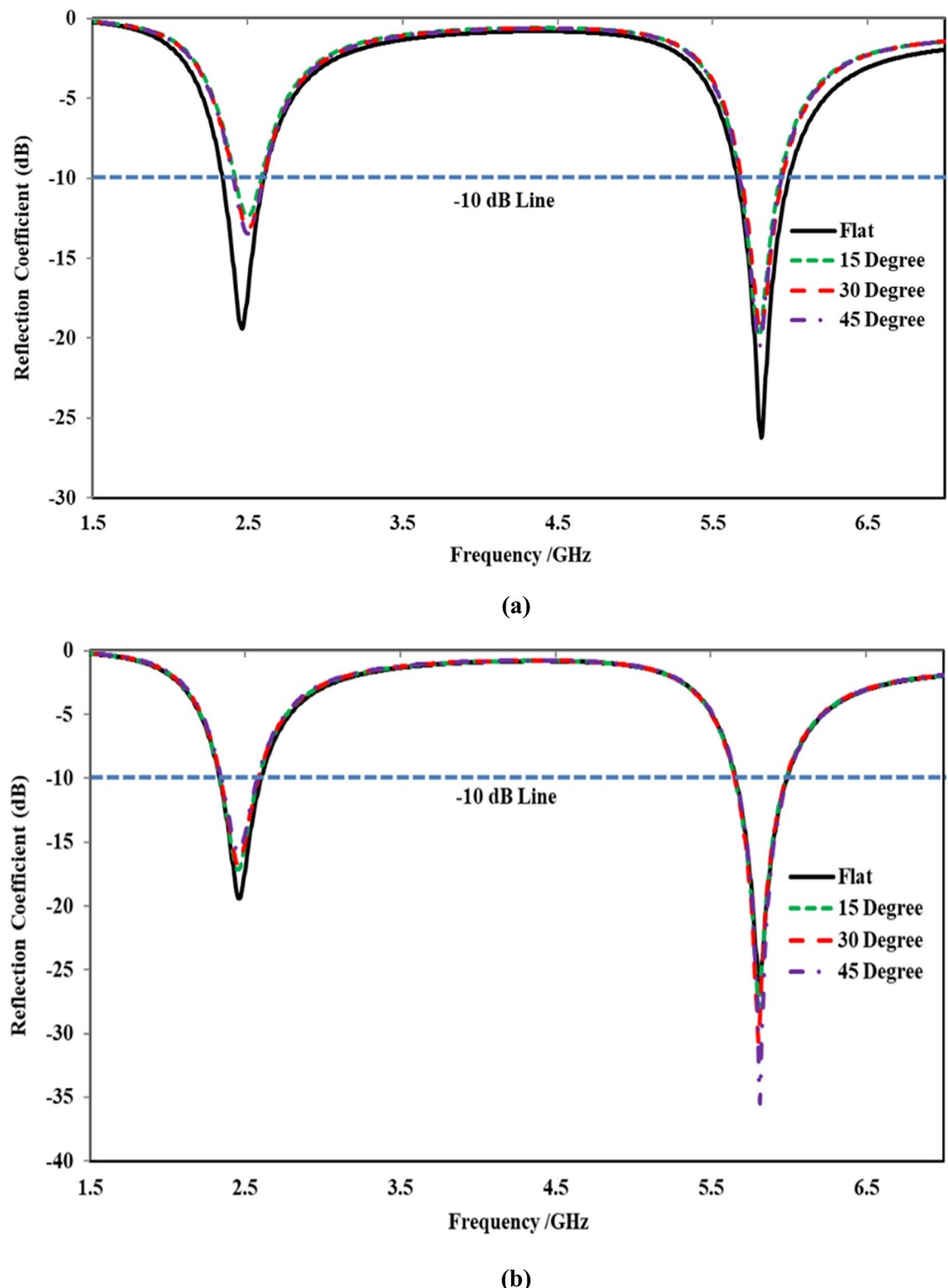

**(a)**

**(b)**

**Fig 14.** Off-body reflection coefficients comparison under flat and bent states in (a) *E*-Plane (b) *H*-Plane.

**Table 3. Performance summary of the proposed antenna under various bending angles.**

| Bending Plane | Angle | Freq. (GHz) | | $S_{11}$ (dB) | | Bandwidth (MHz) | |
|---|---|---|---|---|---|---|---|
| E | 15 Degree | 2.46 | 5.78 | -18.77 | -26.86 | 192 | 284 |
| | 30 Degree | 2.44 | 5.79 | -18.45 | -33.45 | 198 | 295 |
| | 45 Degree | 2.44 | 5.81 | -17.19 | -38.42 | 235 | 310 |
| H | 15 Degree | 2.50 | 5.80 | -12.30 | -19.83 | 169 | 258 |
| | 30 Degree | 2.48 | 5.80 | -13.20 | -19.28 | 191 | 263 |
| | 45 Degree | 2.48 | 5.79 | -13.46 | -20.45 | 196 | 268 |

In the proposed integrated design, a 9 mm ($0.07\lambda_o$) layer of Styrofoam, equivalent to the optimum separation ($x$), is used between the antenna and the AMC surface.

The optimum separation ($x$) between the antenna and the AMC array was determined through a parametric study involving various separation distances between the array and the antenna. The peak gain, reflection coefficient, and radiation efficiency were considered as performance parameters for the parametric study and were analyzed at different values of $x$, such as 3 mm, 6 mm, 9 mm, and 12 mm. The comparison of the reflection coefficient of the proposed antenna at these values of x is illustrated in Fig 16. It was observed that, at all values of $x$, the proposed antenna maintained stability in both frequency bands (i.e., 2.45 GHz and 5.80 GHz). However, the reflection coefficient exhibited a slight shift from the 2.45 GHz central frequency point to the right side at all separation distances, attributable to the size reduction capability of the AMC structure. Additionally, it was noted that increasing the value of $x$ resulted in improved matching. The parametric study further revealed that the antenna's performance in terms of gain and radiation efficiency increased with higher values of $x$. Nevertheless, a separation wider than 9 mm ($0.07\lambda_o$) would increase the overall height of the integrated design. Therefore, a compromise was made between the total height of the integrated design and its performance, and a separation of 9 mm ($0.07\lambda_o$) was considered the optimum distance between the antenna and the AMC array. The performance summary of the proposed integrated design concerning x, including peak gain and radiation efficiency, is presented in Table 4.

The simulated and measured reflection coefficients of the proposed integrated design in the off-body flat state, with the antenna placed at an optimum separation ($x = 0.07\lambda_o$), are shown in Fig 17.

The lower frequency band, centered at 2.45 GHz, resonates with ($|S_{11}|$) < -25 dB, while the upper frequency band, centered at 5.80 GHz, resonates with ($|S_{11}|$) < -23 dB. The measured reflection coefficient reveals that the proposed integrated antenna resonates with ($|S_{11}|$) < -16 dB at 2.45 GHz and ($|S_{11}|$) < -26 dB at 5.80 GHz. Additionally, the simulated -10 dB bandwidth attained from the proposed integrated design at 2.45 GHz and 5.80 GHz is 268 MHz and 293 MHz, respectively. Similarly, the measured -10 dB bandwidth displays a value of 285 MHz and 425 MHz at 2.45 GHz and 5.80 GHz, respectively. The simulated and measured reflection coefficients slightly deviate from each other, which can be attributed to source, human, and fabrication errors. However, the proposed design covers the entire ISM band and is suitable for wearable applications.

The measurement setup for evaluating the far-field gain of the AMC integrated design inside an anechoic chamber is depicted in Fig 18. The far-field gain comparison of the proposed integrated design at the 2.45 GHz and 5.80 GHz frequency bands in both $E$ and $H$-planes is illustrated in Fig 19(A) and 19(B), respectively. It was observed that by employing the AMC surface, the omni-directional radiation transformed into a directional radiation pattern, thereby increasing the peak gain of the antenna in both frequency bands. The simulated gain

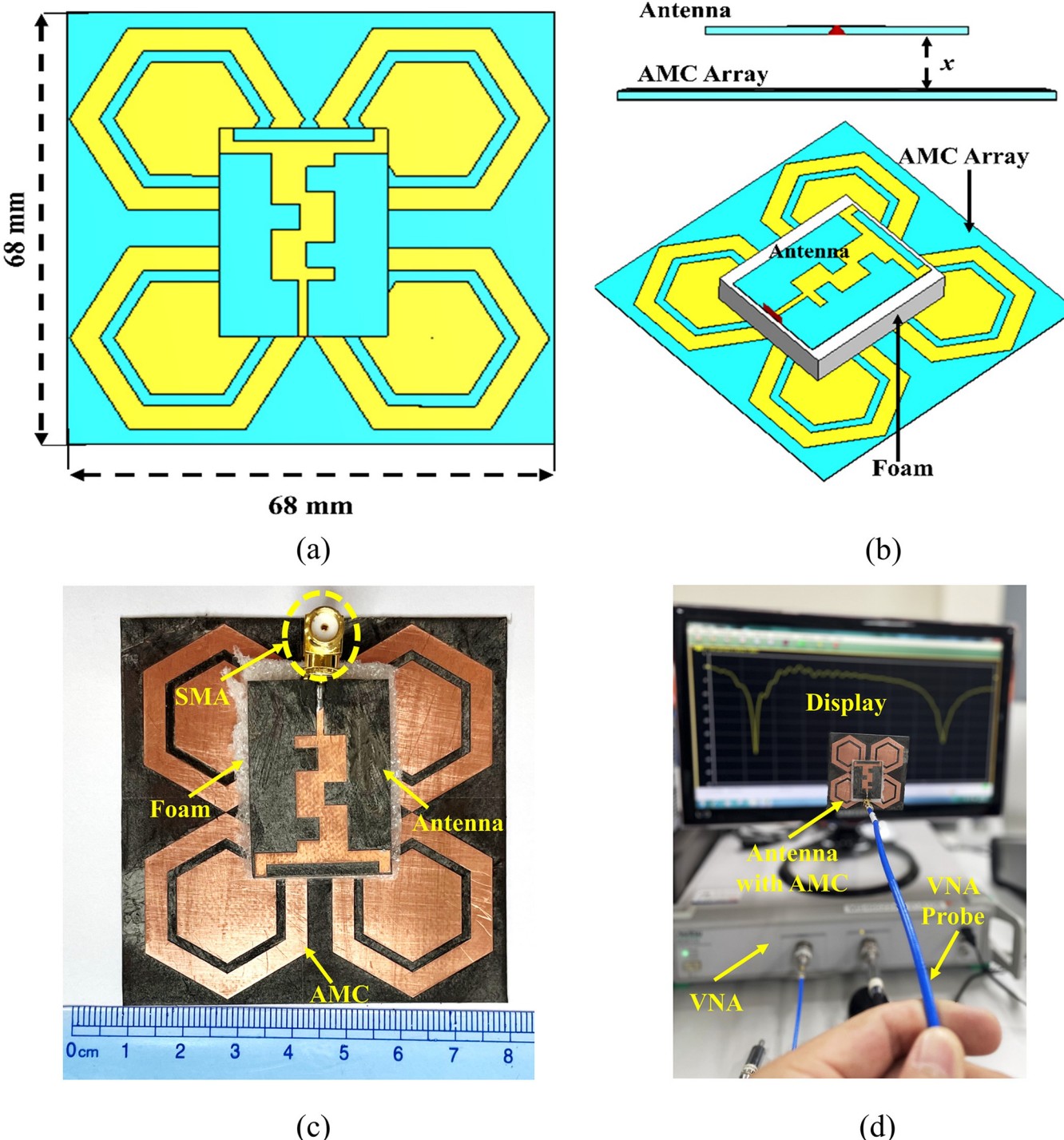

**Fig 15.** Geometry of the 2×2 AMC-integrated design; (a) and (b) CST model; (c) photograph of the fabricated prototype; (d) measurement scenario of the AMC integrated antenna.

at 2.45 GHz when the AMC is employed increases from 1.89 dBi to 6.61 dBi, while at 5.80 GHz, the peak gain enhances from 4.10 dBi to 8.02 dBi. This enhancement in peak gain is attributed to the reduction in back lobe radiation. Similarly, the peak gains were increased

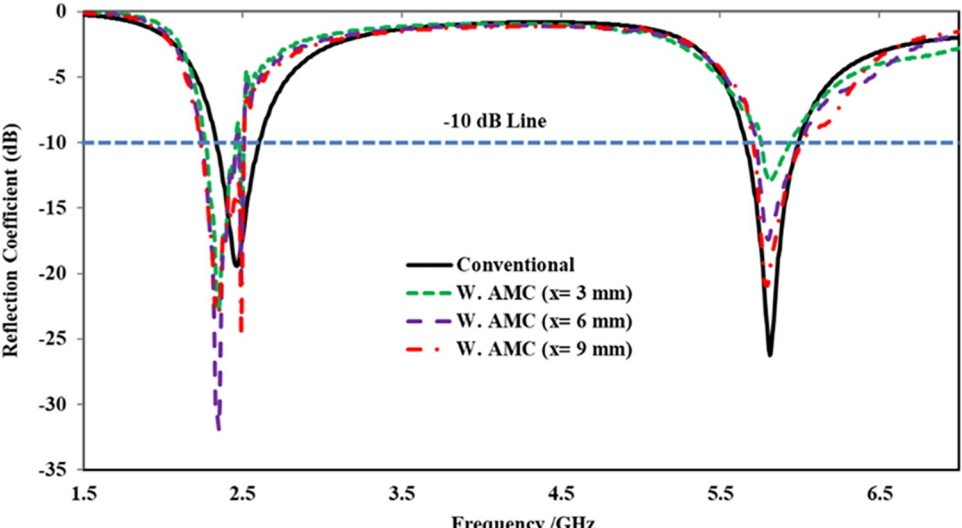

**Fig 16. Simulated reflection coefficient obtained from parametric study with respect to *x* from the AMC array.**

from 2.07 dBi to 6.73 dBi and from 3.12 dBi to 7.79 dBi at 2.45 and 5.80 GHz, respectively. The enhancement in gain at both frequency bands is due to the change in radiation pattern from omnidirectional to directional, which further supports the theory. For clarity, the 3D gain pattern of the proposed AMC integrated design at two distinct frequency bands is illustrated in Fig 20. The off-body performance summary of the proposed antenna with and without AMC integration in a flat state is tabulated in Table 5.

## On-body performance analysis

Body-worn antennas are designed to operate in close proximity to the human body. However, the complex permittivity and conductivity of human body tissues can adversely affect the antenna's overall performance, including its gain, bandwidth, and especially efficiency [49]. Therefore, it is essential to evaluate the antenna's performance in the on-body state following successful performance testing in the off-body state.

To analyze the performance of the antenna, two different types of numerical models—voxel and theoretical models, known as human body phantoms—have been proposed and utilized by antenna engineers [50]. Analyzing the antenna's performance near the human body presents challenges due to the varying properties of human body tissues across frequencies. This complexity is amplified for dual-band wearable antennas. In the proposed approach, to reduce computational cost and complexity, the theoretical tissue model was considered and

**Table 4. Parametic analyis with respect to '*x*' between the AMC arry and the antenna.**

| Distance (*x*) | Gain (dBi) | | Rad. Efficiency (%) | |
|---|---|---|---|---|
| | @ 2.45 GHz | @ 5.80 GHz | @ 2.45 GHz | @ 5.80 GHz |
| 3 mm | 3.22 | 5.31 | 66.19 | 61.22 |
| 6 mm | 4.42 | 6.52 | 75.22 | 71.79 |
| 9 mm | 6.61 | 8.02 | 89.73 | 87.77 |
| 12 mm | 7.23 | 8.82 | 91.88 | 89.81 |

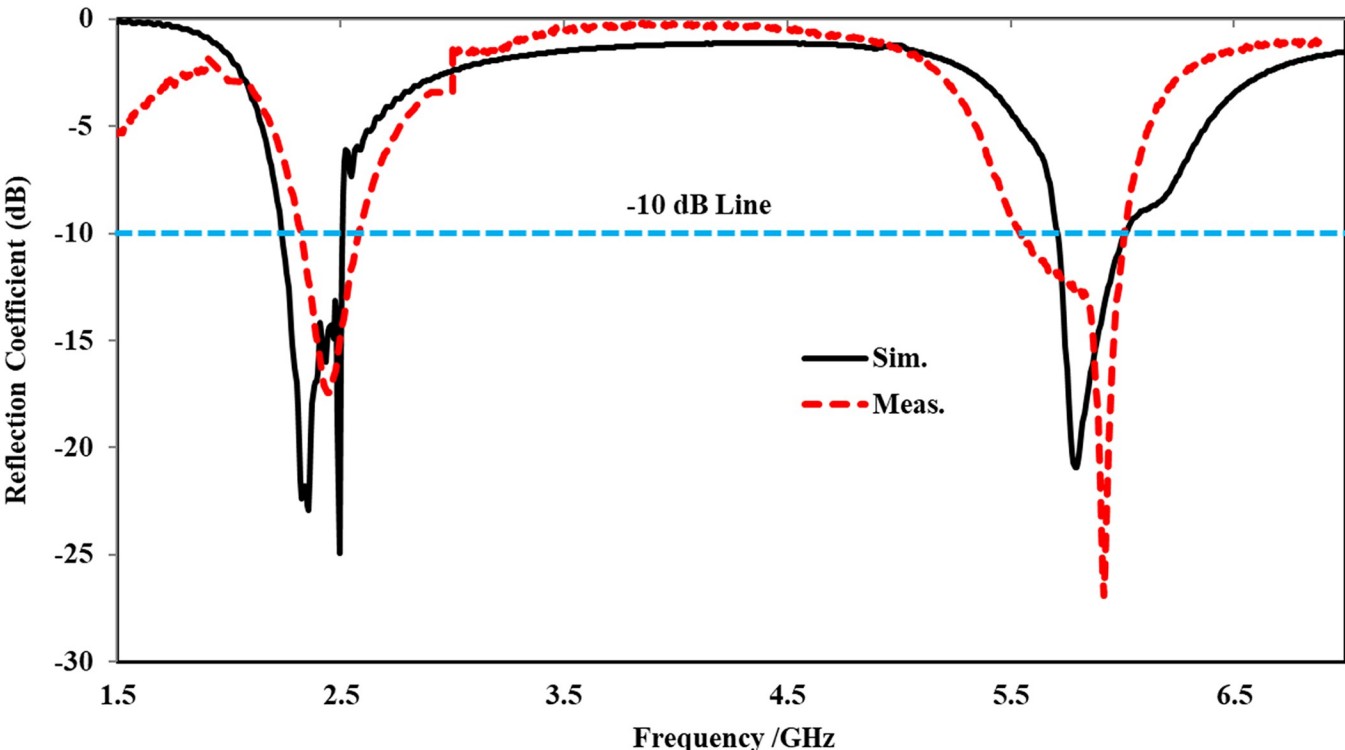

**Fig 17. Reflection coefficient comparison of the proposed AMC integrated antenna at $x = 0.07\lambda_o$.**

implemented it in CST MWS. In this section, we analyzed the antenna's performance, including peak gain, impedance bandwidth, reflection coefficient, and efficiency, both with and without AMC integration, using a three-layer model mimicking the human body chest/back. The model's dimensions were $190 \times 190 \times 21$ mm³, exceeding the surface area of the proposed integrated design. The muscle, fat, and skin layers were 15 mm, 4 mm, and 2 mm thick, respectively, from bottom to top. The electrical properties of these layers are provided in Table 6. To simulate real-world conditions, an air gap and clothing were introduced during simulations, with the proposed designs positioned above the three-layer human body tissue at a distance of $d = 1$ mm.

**Performance analysis without AMC-integration.** In this section, the simulation and testing of the on-body performance of the proposed antenna without integrating AMC are presented and explained. In the simulation, the antenna was placed at a distance $x = 0.07\lambda_o$ above the human body (Fig 21(A)). The fabricated antenna was tested by mounting it at $x = 0.07\lambda_o$ on the human body chest/back of a volunteer, with commercially available Styrofoam used between the antenna and the human body. The measurement setup for the on-body examination is illustrated in Fig 21(B).

The simulated and measured reflection coefficients resulting from on-body evaluation were compared in Fig 22. It was observed that the reflection coefficient of the on-body scenario varied in both frequency bands compared to the antenna operated in the off-body flat state. Furthermore, due to the lossy nature and higher conductivity of body tissues, a slight frequency shift towards the left was noticed in both frequency bands. Additionally, in the higher frequency band centered at 5.80 GHz, the reflection coefficient decreased from -17 dB to -14 dB, resulting in an input impedance of 83 Ω. Compared to the off-body state, the -10 dB

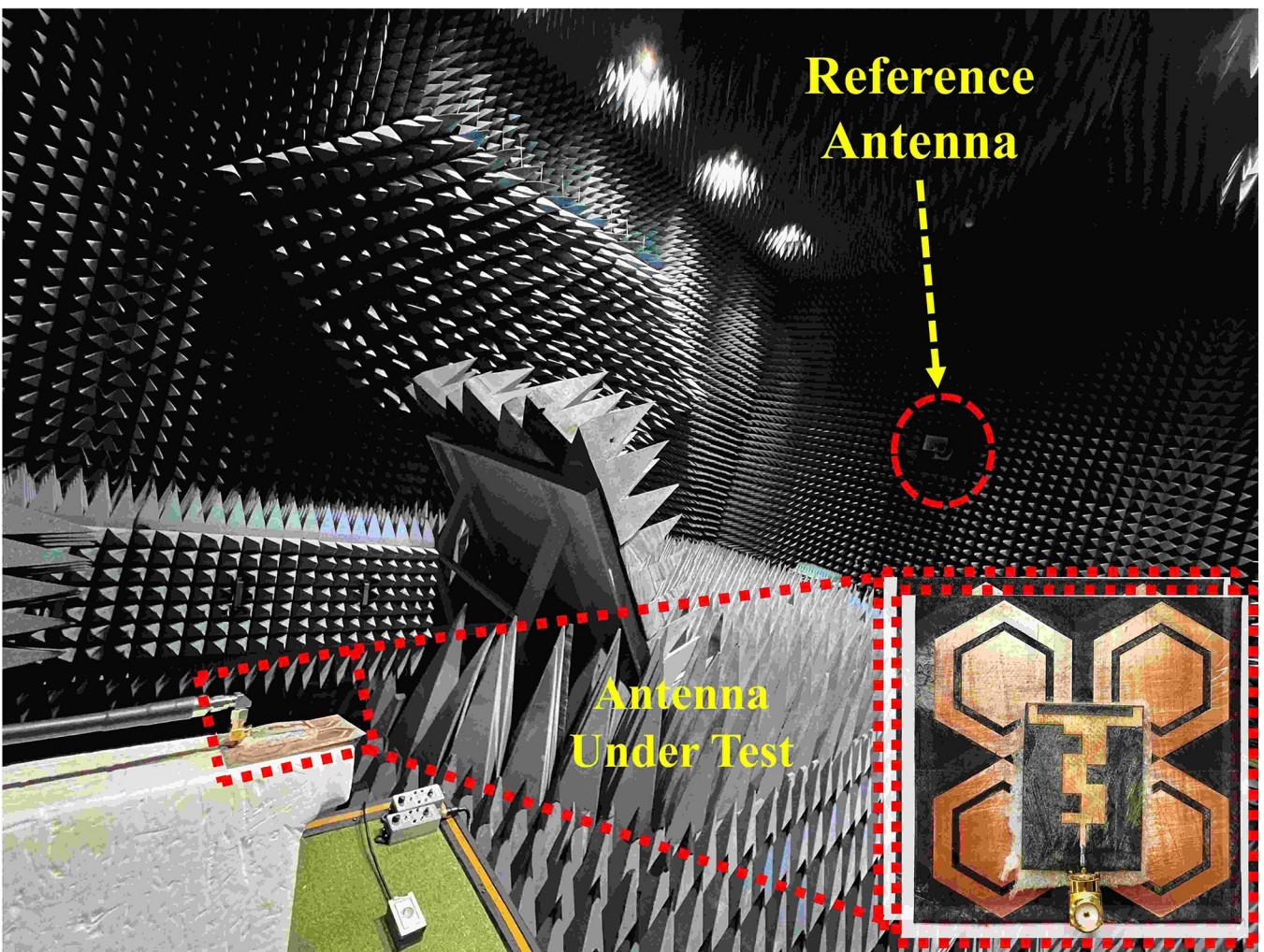

**Fig 18. Far-field gain measurement setup.**

impedance bandwidth of the on-body evaluation was also affected in both frequency bands. In the lower frequency band, the bandwidth was reduced from 279 MHz to 275 MHz. However, at the higher frequency band, this effect was more significant, with the bandwidth reduced from 363 MHz to 99 MHz. The significant decrease in bandwidth at the higher frequency band centered at 5.80 GHz was due to the absorption and scattering of electromagnetic waves in the human body tissue, which was more substantial at 5.80 GHz compared to 2.45 GHz.

The far-field gain comparison of the proposed antenna without integrated AMC in both on- and off-body states in the $E$ and $H$ planes is presented in Fig 23. Testing was conducted in an anechoic chamber by mounting the antenna on a human tissue model. The tissue was modeled with a 190×190×21 mm$^3$ box filled with liquid, comprising 73.2% deionized water, 0.1% NaCl, and 26.7% diethylene glycol butyl ether ($C_8H_{18}O_3$) [53]. Due to the loading of the human body, the radiation characteristics of the antenna deteriorated, resulting in a considerable reduction in peak gain compared to the antenna in the off-body state. At the lower frequency band centered at 2.45 GHz, a significant decrease in peak gain was observed, attributed to the scattering and absorption of electromagnetic waves by the body tissues. The human body absorbs more energy at 2.45 GHz, leading to higher losses and a decrease in the antenna's

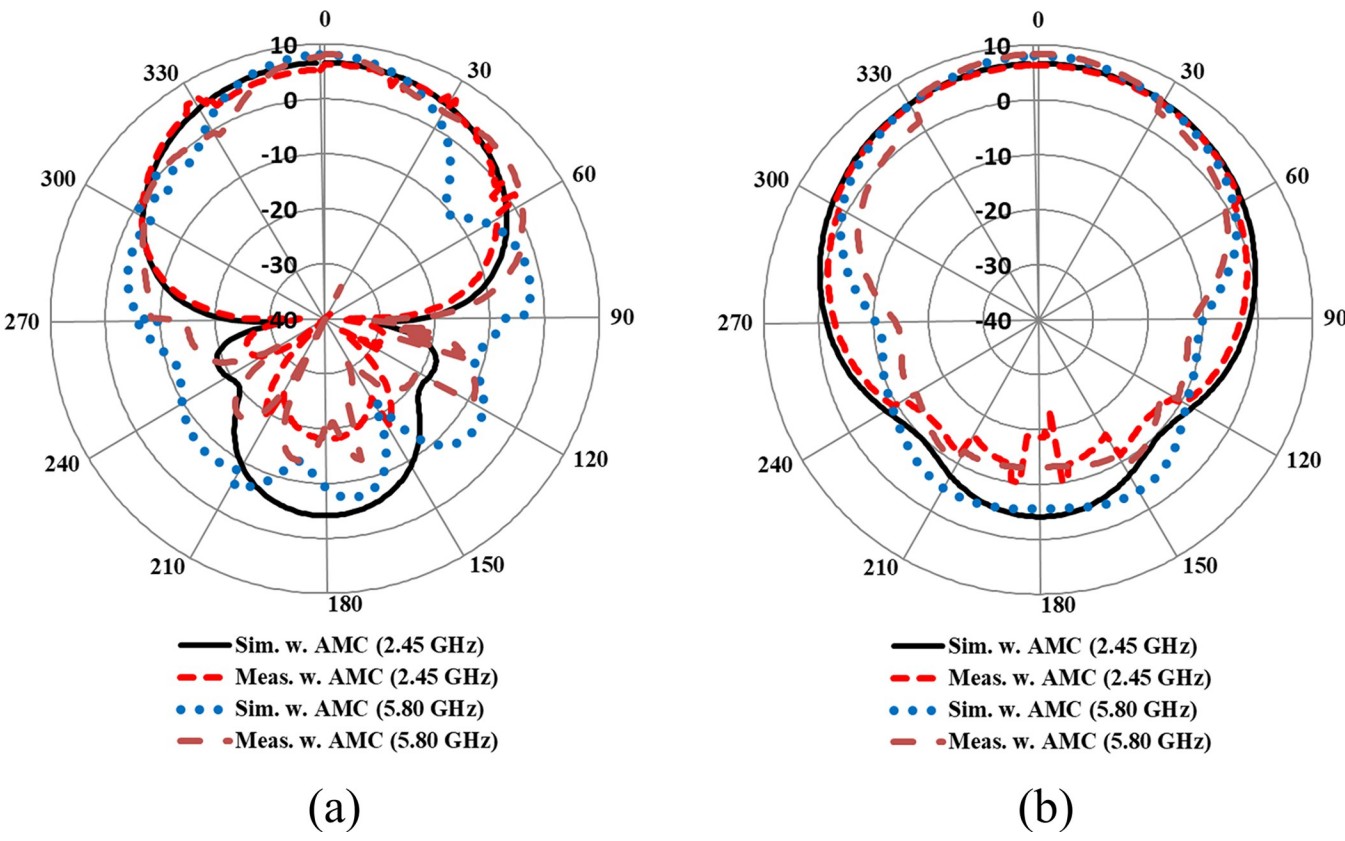

**Fig 19.** Far-field gain comparison of the AMC integrated antenna in (a) *E*-plane (b) *H*-plane.

overall performance. Additionally, the presence of human body tissue alters the distribution of the electromagnetic field, further reducing the peak gain of the antenna at 2.45 GHz. On the other hand, at a higher frequency band centered at 5.8 GHz, the peak gain of the antenna

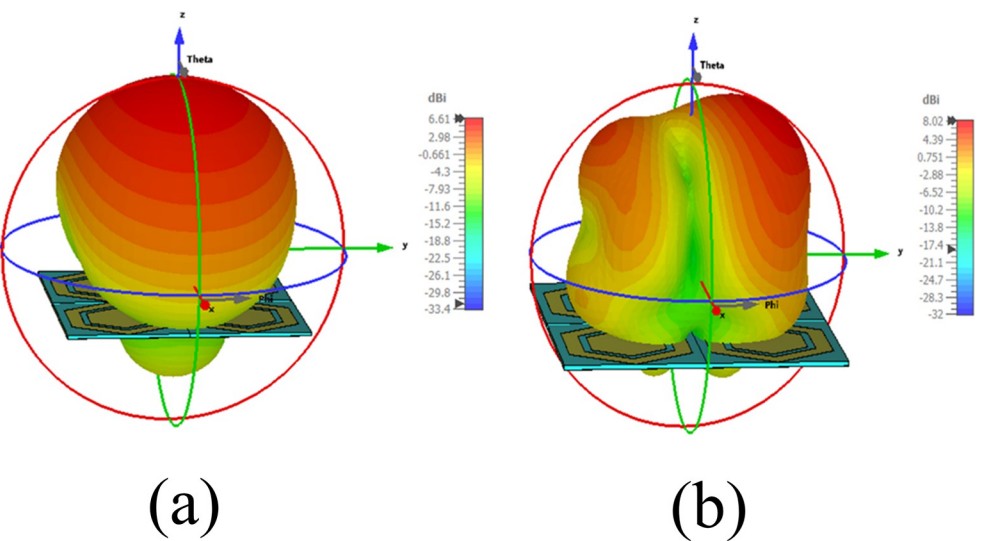

**Fig 20.** 3D gain pattern of the proposed AMC integrated antenna at (a) 2.45 GHz and (b) 5.80 GHz.

**Table 5. Performance summary of the proposed antenna with and without AMC integration.**

| Parameters | Antenna Without AMC | | | | AMC Integrated Antenna | | | |
|---|---|---|---|---|---|---|---|---|
| | @ 2.45 GHz | | @ 5.80 GHz | | @ 2.45 GHz | | @ 5.80 GHz | |
| | Sim. | Meas. | Sim. | Meas. | Sim. | Meas. | Sim. | Meas. |
| Gain (dBi) | 1.89 | 2.07 | 4.10 | 3.12 | 6.61 | 6.73 | 8.02 | 7.79 |
| -10 dB Bandwidth (MHz) | 270 | 279 | 340 | 363 | 268 | 285 | 293 | 425 |
| Radiation efficiency (%) | 96.44 | 87.29 | 92.48 | 84.12 | 89.73 | 86.12 | 87.77 | 83.76 |

slightly increased when placed on a human body. At higher frequencies, the effect of the human body on the electromagnetic field differs from that at lower frequencies. Human body tissues act as reflectors at higher frequencies, resulting in constructive interference and an increase in the antenna's peak gain compared to the off-body state. The summary of the on-body performance of the antenna without integrated AMC is provided in Table 7. The analysis concluded that mounting the antenna on the human body results in most of the power being absorbed by the human body tissues, thereby decreasing the peak gain and radiation efficiency. This also leads to an increase in the value of SAR above specified thresholds, which poses hazards and can have severe health implications.

**Performance analysis with AMC-integration.** As discussed in the previous section, when the antenna was placed in close contact with the human body, the lossy nature of the human body tissues deteriorated the overall performance of the antenna. To mitigate this issue, the AMC surface was employed as a shield to isolate the antenna from the adverse effects of the human body. The AMC surface was positioned at a distance of 1 mm above the human body to account for the air gap and the effect of clothing, while maintaining an optimum separation of $0.07\lambda_o$ between the antenna and the AMC surface. The measurement setup of the proposed integrated design is depicted in Fig 24(B). A 9-mm-thick commercially available Styrofoam was placed between the AMC array and the antenna. The on-body reflection coefficients comparison of the proposed antenna with AMC integration is shown in Fig 25. It was observed that the proposed AMC-integrated antenna exhibited a stable dual-band response compared to the antenna without AMC integration.

Furthermore, the AMC-integrated antenna showed improved simulated and measured impedance matching, with ($|S_{11}|$) < -24 dB at 2.45 GHz and ($|S_{11}|$) < -18 dB at 5.80 GHz. Additionally, the antenna exhibited good input impedance values of 49 Ω and 52 Ω at the 2.45 GHz and 5.80 GHz frequency bands, respectively. The proposed integrated design also provided adequate -10 dB bandwidths in both frequency bands. At the lower frequency band centered at 2.45 GHz, the simulated and measured -10 dB bandwidths were recorded as 269 MHz and 273 MHz, respectively. Similarly, at the higher frequency band centered at 5.80 GHz, the simulated and measured -10 dB bandwidths were recorded as 361 MHz and 365 MHz, respectively. This further demonstrates the contribution of the AMC surface to the antenna's performance when placed in the vicinity of the human body. The tested results are in good agreement with the simulated results, further validating the utility of the proposed AMC

**Table 6. Electrical properties of human body tissues [51, 52].**

| Tissue Layer | Permittivity ($\varepsilon_r$) | Conductivity ($\sigma$) [S/m] | Loss tangent (tan $\delta$) | Permittivity ($\varepsilon_r$) | Conductivity ($\sigma$) [S/m] | Loss tangent (tan $\delta$) |
|---|---|---|---|---|---|---|
| | @ 2.45 GHz | | | @ 5.8 GHz | | |
| Muscle | 52.7 | 1.77 | 0.241 | 48.49 | 4.96 | 0.317 |
| Fat | 5.3 | 0.11 | 0.145 | 4.96 | 0.29 | 0.183 |
| Skin | 38 | 1.46 | 0.282 | 35.12 | 3.71 | 0.328 |

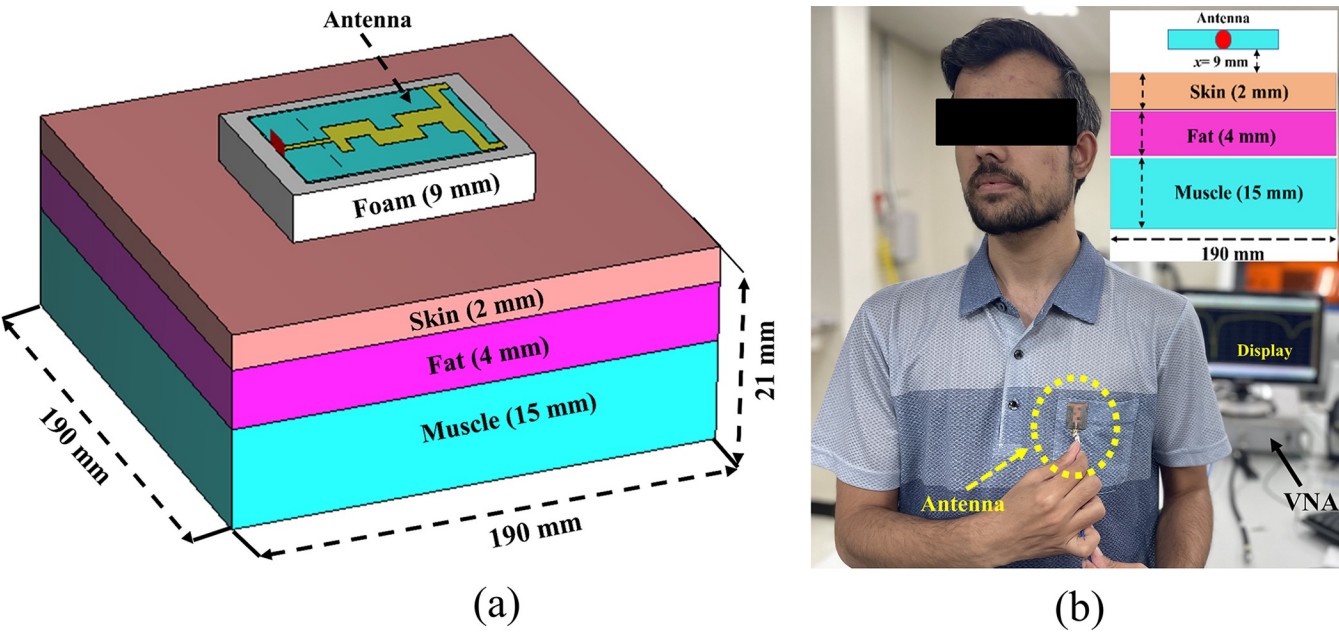

**Fig 21.** Antenna without integrated AMC mounted on a three-layer human body tissue: (a) simulation model, (b) testing scenario.

integrated design in wearable electronics. The on-body simulation and testing of the far-field gain pattern in the *E*-plane of the proposed design with and without integrating AMC at the 2.45 and 5.80 GHz frequency bands are depicted in Fig 26. From Fig 26, it can be observed

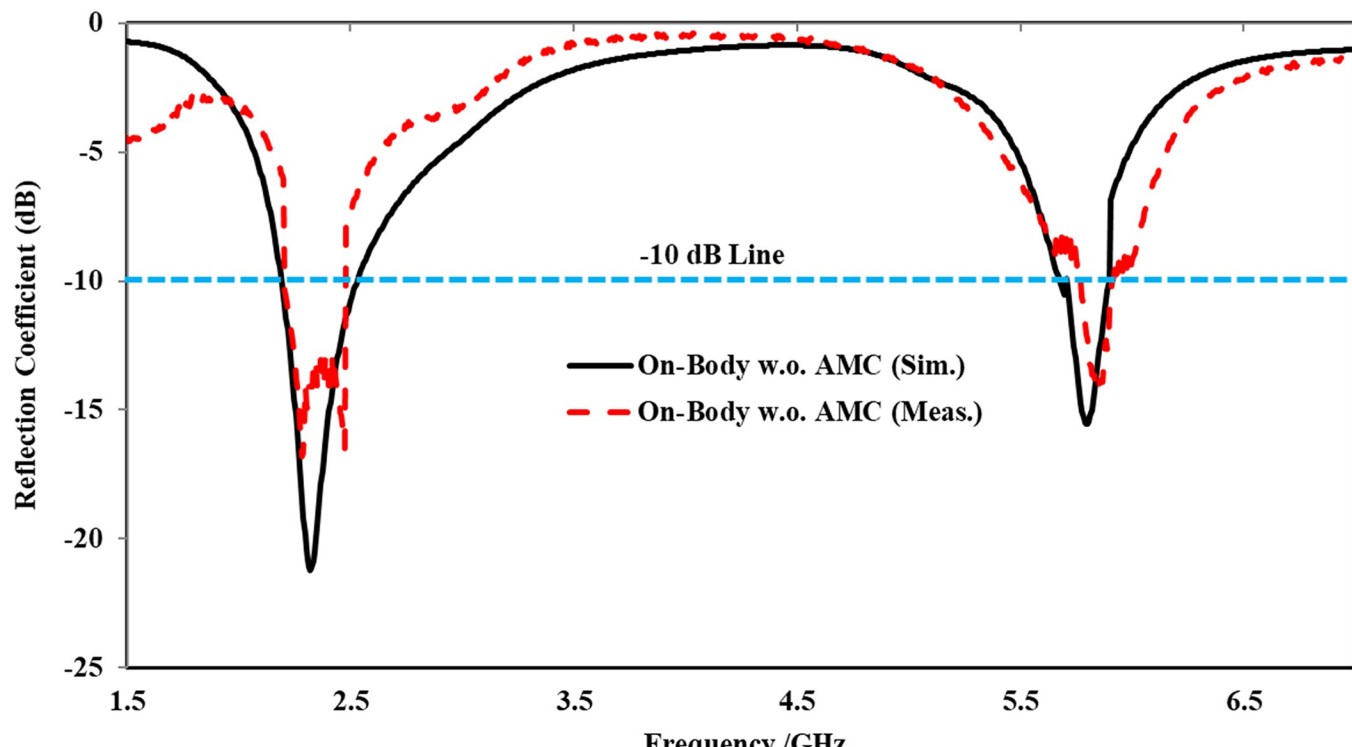

**Fig 22. On-body reflection coefficients comparison without AMC at $x = 0.07\lambda_o$.**

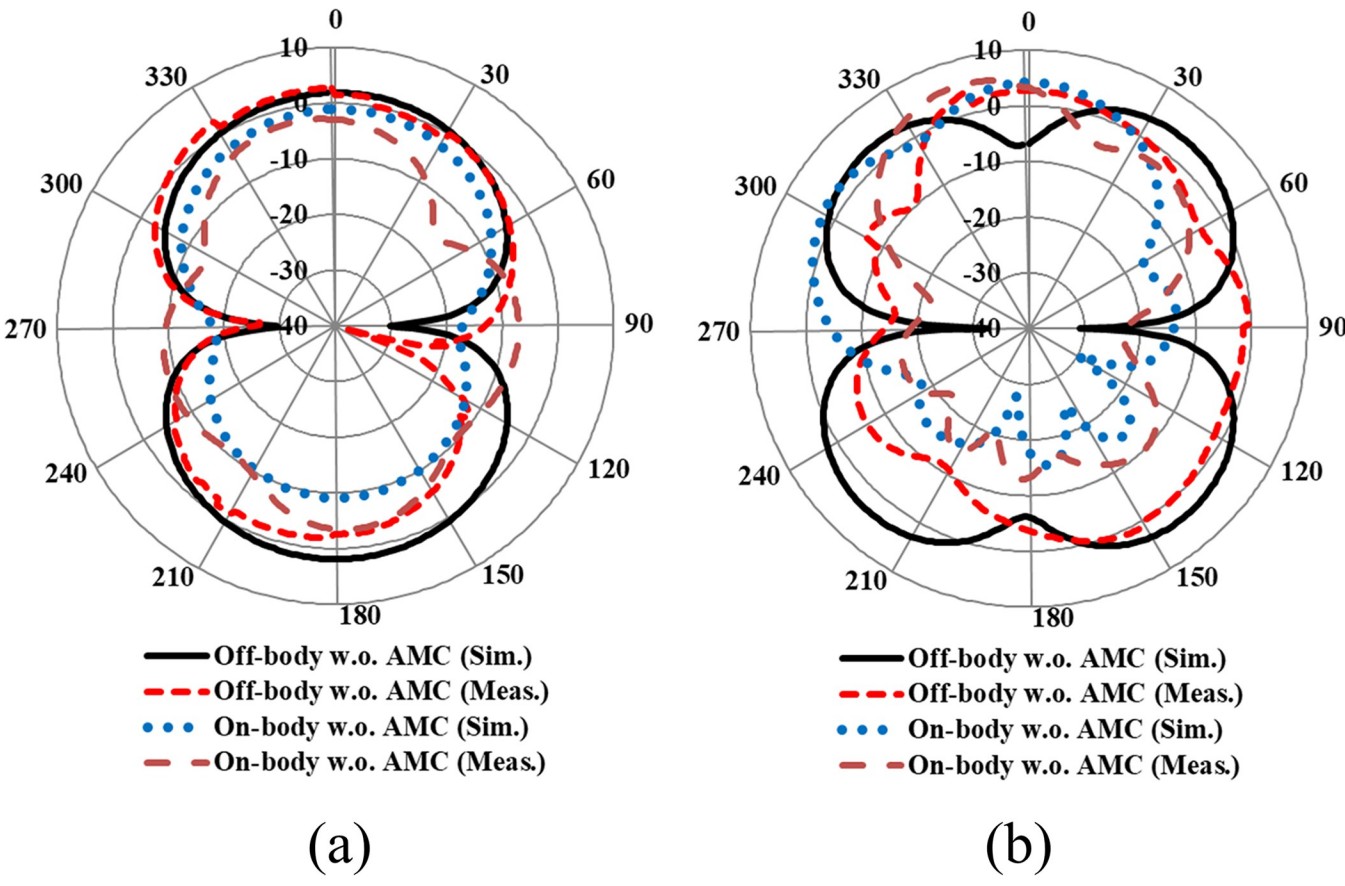

**Fig 23.** On-and off-body gain comparison of the proposed antenna in *E*-plane at (a) 2.45 GHz (b) 5.80 GHz.

that the on-body peak gain of the AMC-integrated design at both frequency bands was greater compared to the antenna without AMC integration, where the peak gain was adversely affected, especially at 2.45 GHz. The peak gain increased from -1.07 dBi to 7.27 dBi at 2.45 GHz and from 4.31 dBi to 6.52 dBi at 5.80 GHz, respectively. For further clarity, a perspective view of the on-body simulated 3D gain pattern of the proposed antenna is provided in Fig 27. The on-body performance of the proposed design integrated with AMC is summarized in Table 7.

## Specific absorption rate (SAR) analysis

As wearable antennas operate in close proximity to the human body, it is necessary to evaluate their Specific Absorption Rate (SAR). SAR limits must not exceed the standard thresholds

**Table 7. On-body performance summary of the proposed antenna.**

| Parameters | Antenna without AMC Integration | | | | Antenna with AMC Integration | | | |
|---|---|---|---|---|---|---|---|---|
| | @ 2.45 GHz | | @ 5.80 GHz | | @ 2.45 GHz | | @ 5.80 GHz | |
| | Sim. | Meas. | Sim. | Meas. | Sim. | Meas. | Sim. | Meas. |
| Gain (dBi) | -1.07 | -1.05 | 4.31 | 4.11 | 7.27 | 7.83 | 6.52 | 4.43 |
| -10 dB Bandwidth (MHz) | 273 | 275 | 103 | 99 | 269 | 273 | 361 | 365 |
| Radiation efficiency (%) | 38.22 | 43.55 | 54.32 | 55.20 | 89.32 | 86.71 | 87.55 | 84.18 |

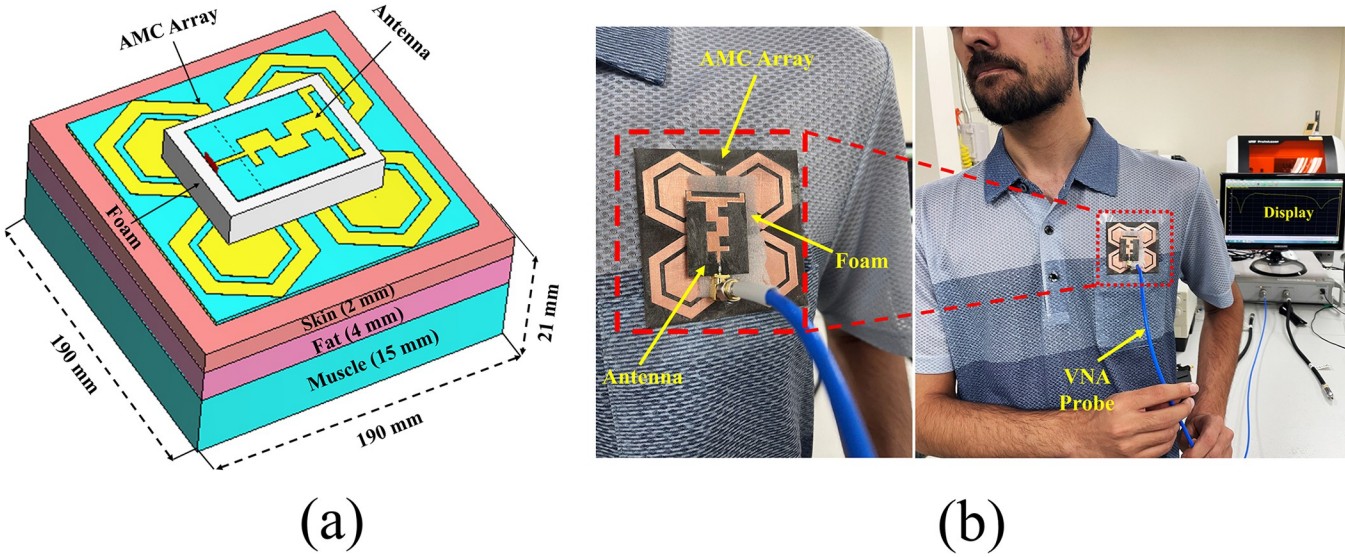

**Fig 24.** On-body analysis of the proposed AMC-integrated antenna placed on a human body phantom (chest): (a) CST model, (b) testing scenario of the proposed design mounted on the volunteer's chest.

defined by the US and EU. The US standard threshold is 1.6 W/kg for any 1g of tissue mass, and the EU standard threshold limit is 2 W/kg for any 10g of tissue mass [54]. The SAR analysis of the proposed designs was conducted using the IEEE C95.1 standard in the CST MWS simulation tool. As a benchmark, an input power of 0.5 watts was considered for computing

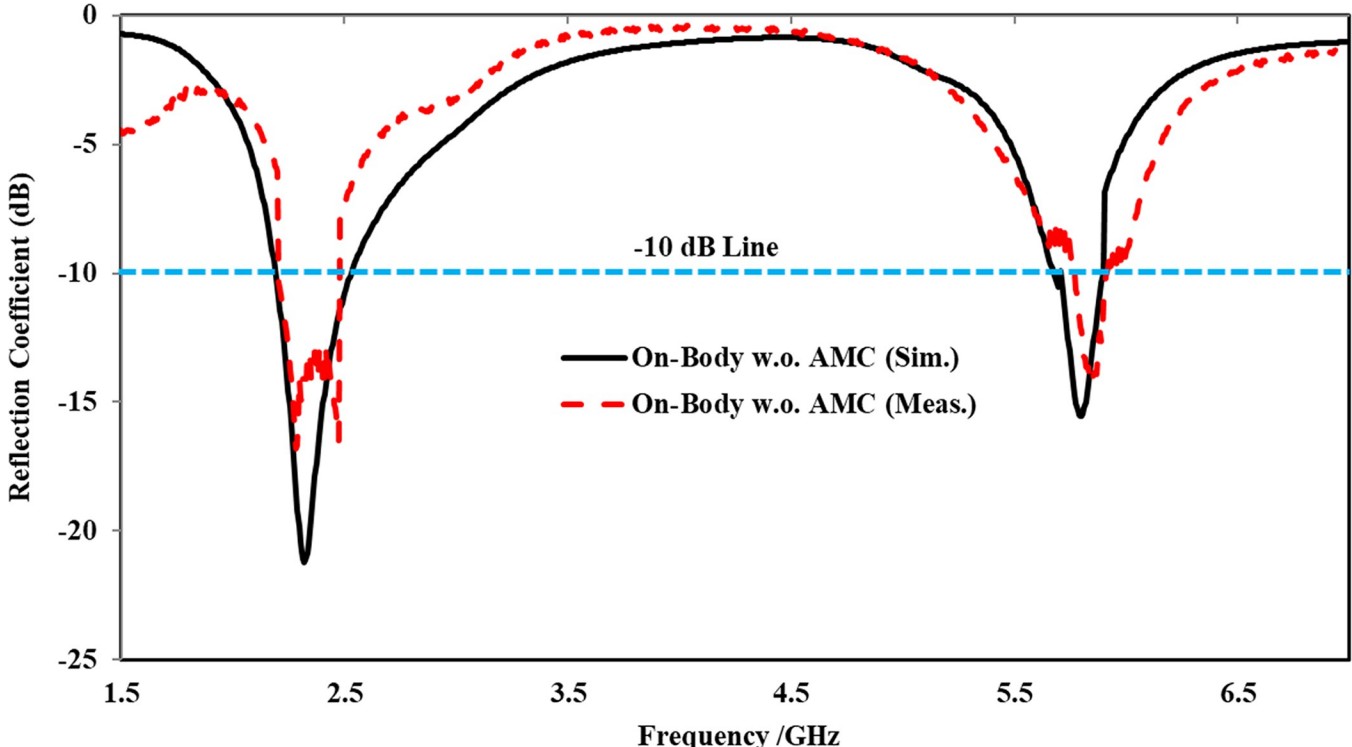

**Fig 25. On-body reflection coefficient comparison of the proposed antenna without and with AMC integration at x = 0.07$\lambda_o$.**

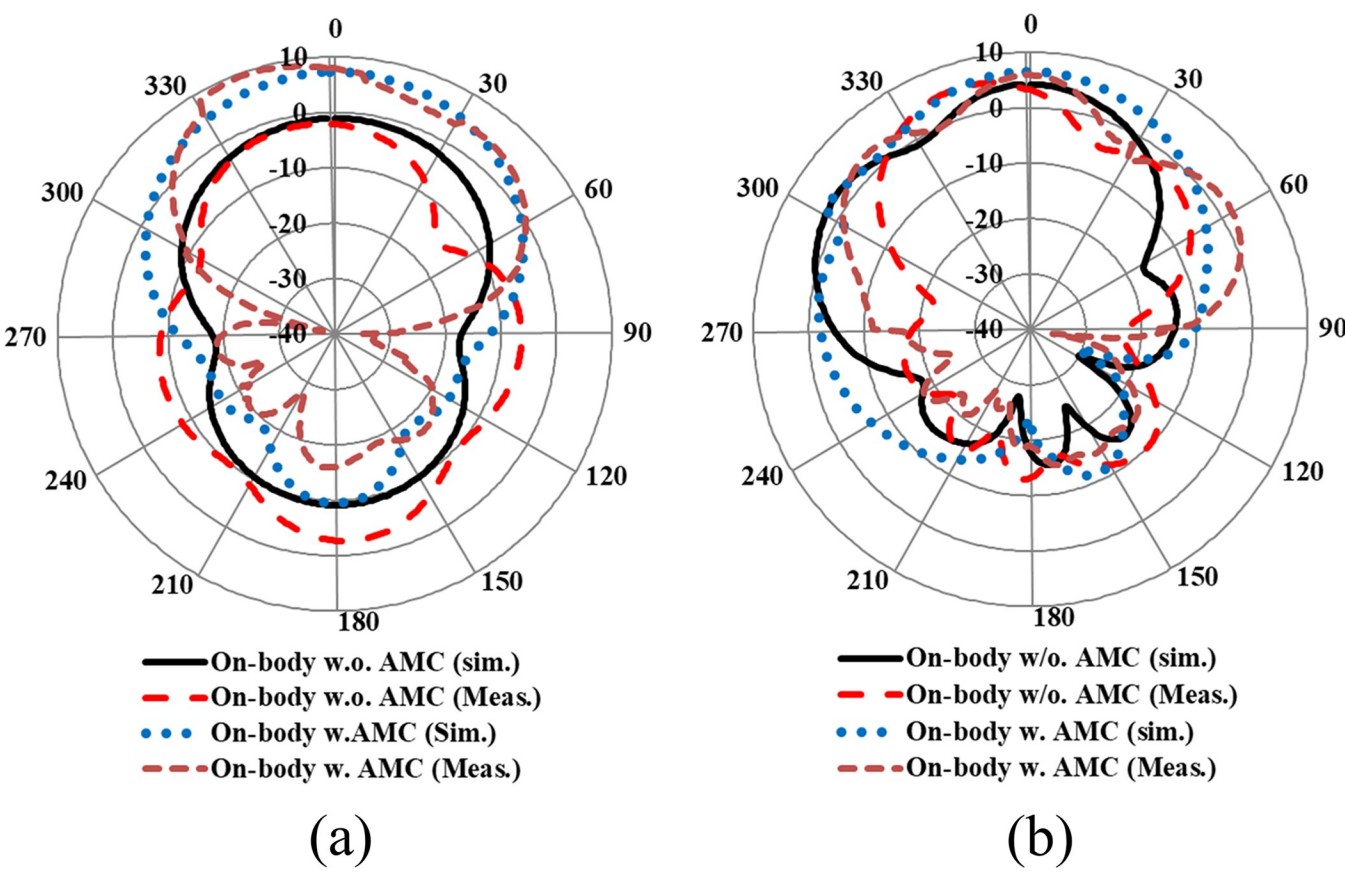

**Fig 26.** On- and off-body gain comparison with and without AMC integration in the *E*-plane at (a) 2.45 GHz and (b) 5.80 GHz.

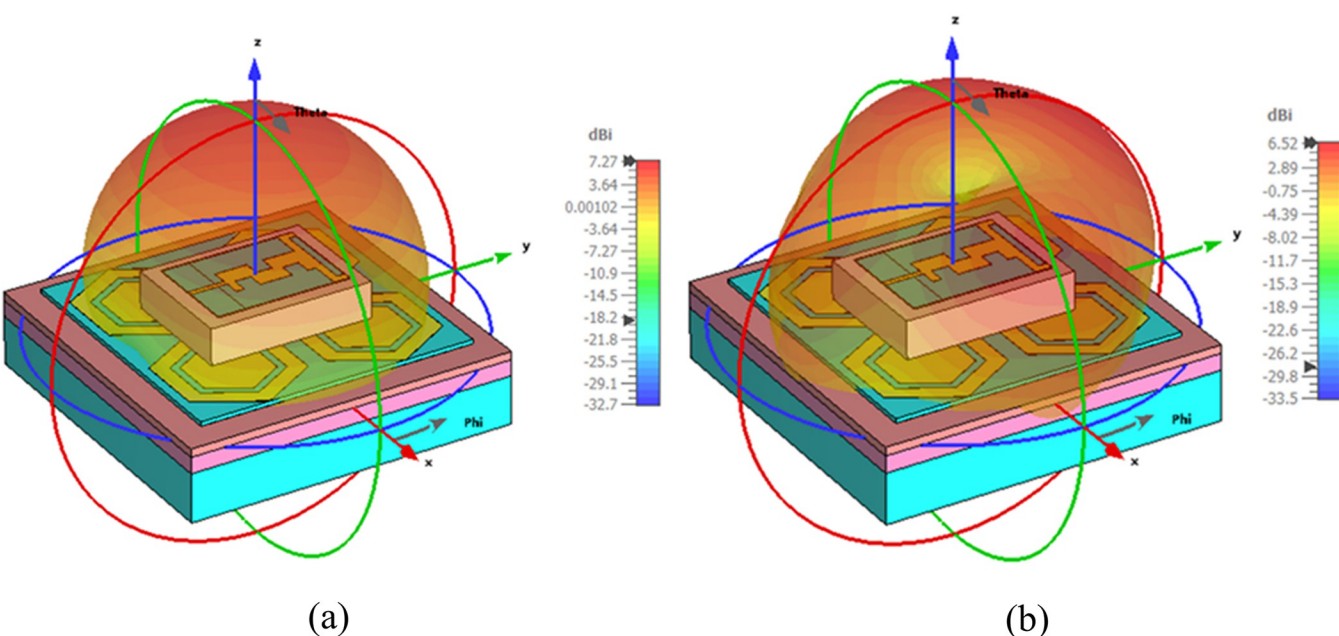

**Fig 27.** On-body 3D gain pattern of the proposed integrated design at (a) 2.45 GHz and (b) 5.80 GHz.

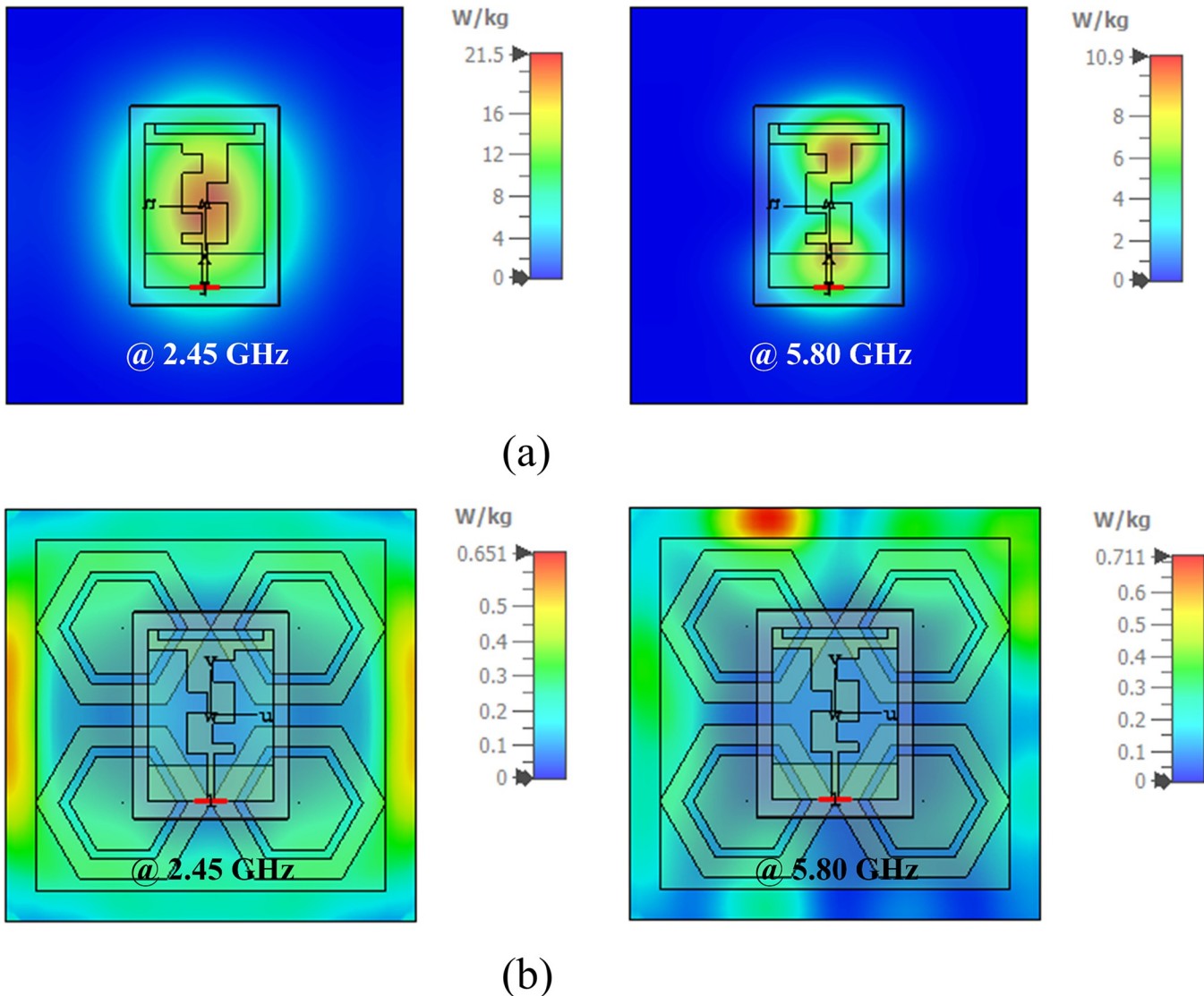

**Fig 28.** SAR distribution of the proposed antenna considering 1g of tissue mass: (a) without AMC integration, (b) with AMC integration (note the different scales).

the SAR value of the antenna with and without AMC integration. The proposed antenna with and without AMC integration was investigated by mounting it at a distance of $0.07\lambda_o$ from a skin layer to account for the thickness of the fabric worn by users and to achieve optimal separation of the antenna from the AMC surface. Fig 28(A) and 28(B) illustrate the simulated SAR distribution of the proposed antenna with and without an integrated AMC structure. When the antenna without AMC integration was placed on the tissue model, the maximum SAR value was observed to be 21.5 and 10.9 W/kg for any 1g of human tissue mass at 2.45 and 5.80 GHz, respectively. These values exceeded the specified limits defined by the US, posing hazards with severe health implications. On the other hand, when the AMC-integrated antenna was placed on human body tissue, the maximum SAR value was observed to be 0.651 and 0.711 W/kg for any 1g of tissue mass at 2.45 and 5.80 GHz, respectively, which was much lower than the specified threshold of 1.6 W/kg for any 1g of tissue. Thus, it was demonstrated that the SAR

**Table 8. Comparative study of the proposed integrated antenna with the existing literature.**

| Ref. | Antenna Size (mm³) | Substrate/ Flexibility | Operating Frequency (GHz) | Back Lobe Reduction Technique | Gain (dBi) | Operating Bandwidth (%age) | Peak SAR (W/Kg) | | Antenna Topology |
|---|---|---|---|---|---|---|---|---|---|
| | | | | | | | 1g | 10g | |
| [54] | (0.33×0.35×0.012) $\lambda_o$ | Rogers/Semi-flexible | 2.4/5.8 | -- | 3.74/ 5.13 | 3.8/5.2 | 0.95/ 0.478 | 0.57/ 0.13 | Planner Monopole |
| [55] | (0.81×0.81×0.026) $\lambda_o$ | F4B/No | 2.45/5.8 | -- | 1.9/5.9 | 2.57/5.22 | 0.254/ 0.074 | -- | Circular |
| [56] | (0.81×0.81×0.016) $\lambda_o$ | Felt/ Yes | 2.45/5.8 | -- | 6.33/ 6.98 | 4.9/3.8 | 0.042/ 0.09 | -- | Circular |
| [57] | (0.98×0.98×0.027) $\lambda_o$ | Textile/Yes | 2.45/5.8 | AMC | 6.4/7.6 | 4/12 | 0.08 | -- | Rec. Patch |
| [58] | (0.82×0.82×0.024) $\lambda_o$ | Textile/ Yes | 2.45/5.5 | AMC | 2.5/0~4 | 12/16.3 | 0.019/ 0.009 | -- | Rec. Patch |
| [59] | (0.67×0.67×0.016) $\lambda_o$ | Felt/ Yes | 2.45/5.8 | EBG | 4.5/5.6 | 4.2 | -- | 0.661/ 1.51 | PIFA |
| [60] | (0.82×0.74×0.024) $\lambda_o$ | Felt/ Yes | 2.4/5.8 | -- | 2.9/5.0 | 6.54/11.5 | -- | 0.056/ 0.067 | SIW cavity antenna |
| [61] | (0.51×0.61×0.022) $\lambda_o$ | Denim/ Yes | 2.45/5.8 | EBG | -- | 4.3/5.57 | -- | 0.031/ 0.040 | Monopole |
| **This Work** | (0.55×0.55×0.002) $\lambda_o$ | Rogers/Semi-flexible | 2.45/5.8 | AMC | 6.61/ 8.02 | 10.12/7.43 | 0.651/ 0.711 | -- | Monopole |

*$\lambda_o$ is the free space wavelength at 2.45 GHz.

limit of the AMC-integrated antenna complies with the safety threshold defined by the US, making the proposed integrated design suitable for wearable applications.

The performance comparison of the proposed integrated antenna with existing state-of-the-art wearable antennas is summarized in Table 8. The antenna in [54], designed on a semi-flexible substrate, is compact but has lower gain and impedance bandwidth compared to the proposed antenna. Additionally, the antenna in [55], fabricated on a rigid substrate, exhibits low gain in the lower frequency band and a relatively narrow operating bandwidth. In contrast, the antenna in [56] demonstrates good gain in both frequency bands but has a limited operating bandwidth compared to the proposed design. When compared to the antennas in [57–60], the proposed design is compact and offers superior performance in terms of gain and impedance bandwidth. The antenna in [61] has a comparable size but lacks information on gain. Furthermore, the proposed integrated antenna meets the low SAR requirements specified by the US at both frequency bands. Due to its good gain, adequate bandwidth, compact size, and lower SAR value, the proposed design is well-suited for body-worn applications.

## Conclusion

In this study, an AMC-based compact dual-band wearable antenna, operating in the 2.45 GHz and 5.80 GHz ISM bands for WBAN applications, was designed and subsequently analyzed. Both the antenna and AMC plane were fabricated on 0.254 mm thick semi-flexible RT/duroid® 5880 material with a relative permittivity of 2.2 and a loss tangent of 0.0009. The total volume of the proposed antenna was 33×24×0.254 mm³, corresponding to $0.26\lambda_o×0.19\lambda_o×0.002\lambda_o$, where $\lambda_o$ is the free space wavelength at 2.45 GHz. To enhance the overall antenna performance and isolate it from the adverse effects of human body tissue, the proposed antenna was backed by a 2×2 AMC array. The total volume of the proposed array was 68×68×0.254 mm³, corresponding to $0.55\lambda_o×0.55\lambda_o×0.002\lambda_o$. The performances of the proposed antenna in both on- and off-body states were analyzed and tested, both with and

without the incorporation of the AMC plane. With the AMC plane, the antenna maintained its performance in both conditions, resulting in a significant increase in peak gain and overall performance across both frequency bands. The proposed AMC-integrated design demonstrated good gain, sufficient bandwidth, and acceptable efficiency in both on- and off-body states. In the off-body state, the design radiated with measured gains of 6.73 dBi and 7.79 dBi, along with a measured bandwidth of 285 MHz and 425 MHz at 2.45 GHz and 5.80 GHz, respectively. Similarly, in the on-body state, measured gains are 7.83 dBi and 4.43 dBi, with a tested bandwidth of 273 MHz and 365 MHz at 2.45 GHz and 5.80 GHz, respectively. As the proposed antenna is to be worn on a human body, bending investigations of the proposed antenna at various angles in the $E$ and $H$-planes were also conducted. It was found that the performance of the proposed design was not affected by bending, ensuring a stable response across all bending angles and states. To assess its suitability for body-worn applications and address safety concerns, SAR investigations were conducted, adhering to FCC standards. The SAR value of the AMC-integrated antenna was reduced from 21.5 W/kg to 0.651 W/kg for any 1g of tissue at 2.45 GHz and from 10.9 W/kg to 0.711 W/kg at 5.80 GHz. These values are lower than the specified limit set by the US, indicating compliance. The proposed integrated antenna is considered safe for WBAN applications. Future directions for this study include further miniaturization of the antenna while maintaining performance, conducting comprehensive investigations on on-body and SAR effects in various body positions and tissue types, exploring advanced metamaterial structures to enhance antenna performance, integrating additional functionalities such as energy harvesting or sensing capabilities, developing adaptive antenna systems for WBAN scenarios, and exploring novel fabrication techniques or materials to improve manufacturability and durability for body-worn applications.

## Acknowledgments

The authors of the manuscript would like to thank to the School of Information and Communications Engineering, Xi'an Jiaotong University, Xi'an, China, for their invaluable contribution to this manuscript. The fabrication and rigorous testing of the antennas presented in this work were made possible through their expertise and facilities.

## Author Contributions

**Investigation:** Usman Ali.

**Methodology:** Usman Ali, Abdul Basir, Babar Kamal.

**Supervision:** Sadiq Ullah.

**Validation:** Abdul Basir, Sen Yan, Hongwei Ren, Babar Kamal.

**Writing – original draft:** Usman Ali, Ladislau Matekovits.

**Writing – review & editing:** Sadiq Ullah, Ladislau Matekovits.

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
