## [Decision Letter · Decision Letter 0]

11 Apr 2024

PONE-D-24-06868Design and Performance Investigation of Metamaterial-Inspired Dual Band Antenna for WBAN ApplicationsPLOS ONE

Dear Dr. Matekovits,

Thank you for submitting your manuscript to PLOS ONE. After careful consideration, we feel that it has merit but does not fully meet PLOS ONE’s publication criteria as it currently stands. Therefore, we invite you to submit a revised version of the manuscript that addresses the points raised during the review process.

We look forward to receiving your revised manuscript.

Kind regards,

Suhaib Ahmed, Ph.D.

Academic Editor

PLOS ONE

2. In the online submission form, you indicated that [available on request].

Additional Editor Comments:

Authors are advised to thoroughly address all the queries raised by the reviewers and make necessary changes in the revised manuscript.

Reviewers' comments:

Reviewer's Responses to Questions

**Comments to the Author**

1. Is the manuscript technically sound, and do the data support the conclusions?

Reviewer #1: Yes

Reviewer #2: Yes

Reviewer #3: Yes

2. Has the statistical analysis been performed appropriately and rigorously? 

Reviewer #1: Yes

Reviewer #2: N/A

Reviewer #3: Yes

3. Have the authors made all data underlying the findings in their manuscript fully available?

Reviewer #1: Yes

Reviewer #2: Yes

Reviewer #3: Yes

4. Is the manuscript presented in an intelligible fashion and written in standard English?

Reviewer #1: Yes

Reviewer #2: Yes

Reviewer #3: Yes

5. Review Comments to the Author

Reviewer #1: From the perspective of research, the authors' work is quite significant and important. My observations are as follows:

(a) The abstract contains the proposed name, objective, and research findings.

(b) The introduction contains the problem statement and the contribution of the work.

(c) The dataset's information is properly cited or well explained, along with justification for the proposed antenna design and suitable outcomes of it.

(d) The proposed outcomes should be compared with existing similar approaches, and the results are presented well in the form of graphs and tables. In this regard, a comparison table is expected. In addition, the authors had to briefly give insights into their work in the form of highlights.

(e) The conclusion contains the findings of the work and the future scope.

The rest of it looks good, in my opinion.

Reviewer #2: The authors have reported the work title “Design and Performance Investigation of Metamaterial-Inspired Dual Band Antenna for WBAN Applications,”. However, there are few queries which need to be addressed by the authors and are given below

Q1. Why Rogers RTDuroid5880 Substrate is termed as semi-flexible in abstract?

Q2. What do you mean by in-phase characterization?

Q3. Add more analysis on AMC

Q4. Give more insight and clear image on surface current density distribution on the surface?

Q5. The slight deviation in S11 for bending analysis is observed. Give reasons.

Q6. Why there are changes in S11 result when distance between antenna and AMC is varied?

Q7. Why foam is preferred between Antenna and AMC?

Q8. Can you add mathematical modelling of AMC

Q9. Add back-lobe reduction technique comparison in Table 8.

Reviewer #3: Dear authors;

1-Can you summarize in a few sentences what all this work is about at the end of the introduction?

2-Theabstract needs more interest and rewriting of some paragraphs.

3-There are still some aspects that can be improved (for grammar and punctuation). Improve the technical writing of your paper, where there are several grammatical errors and spelling, I think they need to be checked out.

4-The conclusion needs more effort to elaborate on the achieved results with respect to future work,

5-There are still some aspects regarding the obtained results discussions that are missing. Can you please address your achievements well?

6-The practical part is very important; therefore, I’m asking about the application specifications and research gap,

7- The proposed topic is interesting, However, minor variations on the investigated subject have been noted.

8- The research method is not clearly described and is only partially appropriate for addressing the identified problem.

9- The circuit model is very important?

10- The problem statement is unclear, and the literature review is articulated unclearly.

11- The Results and Discussion section requires extension. The findings presented require a detailed justification and careful comparative analysis with previously published works to show their relevance and contribution to the field.

12- The future impact of this work is not clearly defined in the content, limiting the impact of this work. Future work is an important part of the conclusion.

13- The results are still not matured well. Can you clearly mention what your objectives clearly for each result you got during the discussion process?

14- The references need to be written in the same format.

15- There are several relative works, I wish to discuss. Can you please suggest your enhancements and advancements (if you would) over such as the following:

A- https://doi.org/10.1049/iet-nde.2020.0013

B- http://dx.doi.org/10.2528/PIERM20113008

C- https://doi.org/10.1002/dac.5024

D- https://doi.org/10.1002/mop.32067

E- https://doi.org/10.1007/s11277-020-07646-y

F- https://doi.org/10.1017/S1759078720000665

G- https://doi.org/10.1016/j.aeue.2010.03.008

I loved this work and I feel it is very good. I hope these comments will help you improve this work after a major revision.

Regards

6. PLOS authors have the option to publish the peer review history of their article (what does this mean?). If published, this will include your full peer review and any attached files.

Reviewer #1: No

Reviewer #2: No

Reviewer #3: No

---

## [Decision Letter · Decision Letter 1]

23 Jun 2024

Design and Performance Investigation of Metamaterial-Inspired Dual Band Antenna for WBAN Applications

PONE-D-24-06868R1

Dear Dr. Matekovits,

We’re pleased to inform you that your manuscript has been judged scientifically suitable for publication and will be formally accepted for publication once it meets all outstanding technical requirements.

Kind regards,

Suhaib Ahmed, Ph.D.

Academic Editor

PLOS ONE

Additional Editor Comments (optional):

Thank you for addressing all the raised queries in the previous review round. Reviewers have accepted the revised manuscript in its current form.

Reviewers' comments:

Reviewer's Responses to Questions

**Comments to the Author**

1. If the authors have adequately addressed your comments raised in a previous round of review and you feel that this manuscript is now acceptable for publication, you may indicate that here to bypass the “Comments to the Author” section, enter your conflict of interest statement in the “Confidential to Editor” section, and submit your "Accept" recommendation.

Reviewer #2: All comments have been addressed

Reviewer #3: All comments have been addressed

2. Is the manuscript technically sound, and do the data support the conclusions?

Reviewer #2: Yes

Reviewer #3: Yes

3. Has the statistical analysis been performed appropriately and rigorously? 

Reviewer #2: Yes

Reviewer #3: Yes

4. Have the authors made all data underlying the findings in their manuscript fully available?

Reviewer #2: Yes

Reviewer #3: Yes

5. Is the manuscript presented in an intelligible fashion and written in standard English?

Reviewer #2: Yes

Reviewer #3: Yes

6. Review Comments to the Author

Reviewer #2: Authors have incorporated all the comments. The paper is accepted in current form. They have answered all the queris which is technically supported by proof.

Reviewer #3: I have no further comments the authors did everything well. All comments are addressed well with all corrections.

7. PLOS authors have the option to publish the peer review history of their article (what does this mean?). If published, this will include your full peer review and any attached files.

Reviewer #2: **Yes: **Manish Sharma

Reviewer #3: No

---

## [Editor Report · Acceptance letter]

2 Jul 2024

PONE-D-24-06868R1 

PLOS ONE

Dear Dr. Matekovits, 

I'm pleased to inform you that your manuscript has been deemed suitable for publication in PLOS ONE. Congratulations! Your manuscript is now being handed over to our production team.

Kind regards, 

on behalf of

Dr. Suhaib Ahmed 

Academic Editor

PLOS ONE